# The influence of model spatial resolution on simulated ozone and fine particulate matter for Europe: implications for health impact assessments

Sara Fenech[1,2], Ruth M. Doherty[1], Clare Heaviside[2], Sotiris Vardoulakis[3], Helen L. Macintyre[2], Fiona M. O'Connor[4]

[1] School of GeoSciences, University of Edinburgh

[2] Centre for Radiation, Chemical and Environmental Hazards, Public Health England

[3] Institute of Occupational Medicine

[4] Met Office, Hadley Centre, Exeter, UK

*Correspondence to*: Sara Fenech (S.Fenech@sms.ed.ac.uk)

**Abstract.** We examine the impact of model horizontal resolution on simulated surface ozone ($O_3$) and particulate matter less than 2.5 μm ($PM_{2.5}$) concentrations, and the associated health impacts over Europe, using the HadGEM3-UKCA chemistry-climate model to simulate pollutant concentrations over Europe at a coarse (~ 140 km) and a finer (~ 50 km) resolution. The attributable fraction (AF) of total mortality due to long-term exposure to warm season daily maximum 8-hour running mean (MDA8) $O_3$ and annual-average $PM_{2.5}$ concentrations is then calculated for each European country using pollutant concentrations simulated at each resolution. Our results highlight a seasonal variation in simulated $O_3$ and $PM_{2.5}$ differences between the two model resolutions in Europe. Compared to the finer resolution results, simulated European $O_3$ concentrations at the coarse resolution are on average higher in winter and spring (10% and 6%, respectively). In contrast, simulated $O_3$ concentrations at the coarse resolution are lower in summer and autumn (-1% and -4%, respectively). These differences may partly be explained by differences in nitrogen dioxide ($NO_2$) concentrations simulated at the two resolutions. Compared to $O_3$, we find the opposite seasonality in simulated $PM_{2.5}$ differences between the two resolutions. In winter and spring, simulated $PM_{2.5}$ concentrations are lower at the coarse compared to the finer resolution (-8% and -6%, respectively) but higher in summer and autumn (29% and 8%, respectively) and are mostly related to differences in convective rainfall between the two resolutions for all seasons. These differences between the two resolutions exhibit clear spatial patterns for both pollutants that vary by season, and exert a strong influence on country to country variations in estimated AF for the two resolutions. Warm season MDA8 $O_3$ levels are higher in most of southern Europe, but lower in areas of northern and eastern Europe when simulated at the coarse resolution compared to the finer resolution. Annual-average $PM_{2.5}$ concentrations are higher across most of northern and eastern Europe but lower over parts of southwest Europe at the coarse compared to the finer resolution. Across Europe, differences in the AF associated with long-term exposure to population-weighted MDA8 $O_3$ range between -0.9 % and +2.6 % (largest positive differences in southern Europe) while differences in the AF associated with long-term exposure to

population-weighted annual mean $PM_{2.5}$ range from -4.7% to +2.8% (largest positive differences in eastern Europe) of the total mortality. Therefore this study, with its unique focus on Europe, demonstrates that health impact assessments calculated using modelled pollutant concentrations, are sensitive to a change in model resolution by up to ±5% of the total mortality across Europe.

## 35   1 Introduction

A substantial number of epidemiological studies have derived risk estimates for mortality associated with long-term exposure to ambient fine particulate matter with aerodynamic diameter less than 2.5 µm ($PM_{2.5}$) ( Krewski, 2009;Brook et al., 2010; WHO, 2013) and also recently, to a lesser extent, for long-term exposure to ozone ($O_3$) ( Jerrett et al., 2009; Forouzanfar et al., 2016; Turner et al., 2016). Differences in risk estimates produced from different epidemiological studies can be due to

differences in methodologies, air pollution and health data used including the size and spatial extent of cohort populations. For $O_3$, these long-term risk estimates are derived from North American studies. In this region $O_3$ data is typically monitored only during the $O_3$ season (April-September), hence these derived $O_3$-risk estimates apply only to the ozone occurring in the warm season part of the year.

Air pollutant exposure estimated from concentrations measured at fixed monitoring stations, is often used to estimate

health impacts at the cohort-scale (Cohen et al., 2004). However, quantifying the adverse health effects of air pollution at the continental-scale requires atmospheric models (with resolutions ranging from ~250 to 50 km) to simulate pollutant spatio-temporal distributions across these scales (e.g. West et al. 2009; Anenberg et al. 2010; Fang et al. 2013; Silva et al. 2013; Lelieveld et al. 2015, Malley et al., 2017). Amongst a number of factors, simulated air pollutant concentrations may vary depending on the three-dimensional chemistry model used, its set-up and the model resolution (e.g. Markakis et al. 2015;

Schaap et al. 2015; Yu et al. 2016; Neal et al. 2017). Although the same model processes are represented at different model resolutions, simulated pollutant concentrations can vary due to differences in (i) the resolution of emissions, which may have a nonlinear effect on the chemical formation of pollutants, and (ii) the resolution of the driving meteorology (Valari and Menut 2008; Tie et al. 2010; Arunachalam et al. 2011; Colette et al. 2013; Markakis et al. 2015; Schaap et al. 2015).

The impact of model horizontal resolution on simulated $O_3$ concentrations has been primarily linked to less dilution

of emissions when using a finer resolution (Valari and Menut 2008; Tie et al. 2010; Colette et al. 2013; Stock et al. 2014; Schaap et al. 2015). Investigating the impact of increasing model horizontal resolution from 48 km to 6 km on $O_3$ concentrations in Paris, Valari and Menut (2008) found modelled surface $O_3$ to be more sensitive to the resolution of input emissions than to meteorology. A number of other studies note the sensitivity of simulated $O_3$ to simulated nitrogen oxide ($NO_x$) concentrations that determine the extent of titration of $O_3$ by nitrogen monoxide (NO) (Stock et al., 2014; Markakis et

al., 2015; Schaap et al., 2015). Furthermore, Stock et al. (2014) found the impact of spatial resolution (150km vs. 40km) on simulated $O_3$ concentrations to vary with season across Europe. In winter, higher $NO_x$ concentrations produced more pronounced titration effects on $O_3$ at 40 km resolution with a mean bias error (MBE) of 3.2%, leading to lower $O_3$

concentrations than at 150 km resolution (MBE = 14.4%). In summer, although similar results were found for $O_3$ concentrations simulated at the coarse (MBE = 29.7%) and fine resolution (MBE = 32.8%) simulated boundary layer height was suggested to be largely responsible for the spatial differences in $O_3$ concentrations at the two resolutions.

PM$_{2.5}$ concentrations have also been found to be sensitive to the model horizontal resolution (Arunachalam et al. 2011; Punger and West 2013; Markakis et al. 2015; Neal et al. 2017). In the U.S., Punger and West (2013) found population-weighted annual mean PM$_{2.5}$ concentrations to be 6% higher at 36 km compared to 12 km, but 27% lower when simulated at 408 km compared to 12 km. However in this study, statistical averaging was used to estimate pollutant concentrations at the coarsest resolutions, and therefore differences in emissions and meteorology and their atmospheric processing between the resolutions were not included. In contrast, Li et al. (2015) found annual mean PM$_{2.5}$ concentrations simulated at a resolution of ~ 2.5° in the U.S. to be similar to PM$_{2.5}$ concentrations simulated at a resolution of ~ 0.5° suggesting that the horizontal scales being compared and the methodology for comparison are important. However maximum PM$_{2.5}$ concentrations which occur in highly populated regions were found to be 21% lower at the coarse resolution (Li et al., 2015).

As outlined above, a number of studies have analysed the effect of model resolution on $O_3$ and PM$_{2.5}$ concentrations but few have looked at the sensitivity of the associated health impacts to model resolution (Punger and West 2013; Thompson et al. 2014; Li et al. 2015). Punger and West (2013) found mortality associated with long-term exposure to $O_3$ in the US to be 12% higher at a 36 km resolution compared to the mortality estimate at 12 km, as a result of higher $O_3$ simulated at the coarser-scale. Thompson et al. (2014) also found that especially in urban areas, the human health impacts associated with differences in $O_3$ between 2005 and 2014 calculated using a coarse resolution model (36 km) were on average two times greater than those estimated using finer scale resolutions (12 km and 4 km). In addition, Thompson and Selin (2012) found that the estimated avoided $O_3$-related mortalities between a 2006 base case and a 2018 control policy scenario at a 36 km resolution were higher compared to estimates at the finer resolutions (12 km , 4 km and 2 km) . However, their health estimates at the 36 km resolution fall within the range of values obtained using concentrations simulated at the finer resolutions used.

For PM$_{2.5}$-related health estimates, studies by Punger and West (2013) and Li et al. (2015) both found that attributable mortality associated with long-term exposure to PM$_{2.5}$ in the US was lower for their coarser resolution simulations (> 100 km) due to lower simulated PM$_{2.5}$ concentrations in densely populated regions. However, Thompson et al. (2014) found that using model horizontal resolutions of 36, 12 and 4 km had a negligible effect on changes in PM$_{2.5}$ concentrations and associated health impacts. This is likely due to the relatively small range of resolutions used by Thompson et al. (2014) compared to these other studies.

The majority of health effect studies relating to the impact of model resolution have been conducted in North America. Hence, similar studies are lacking over Europe. This study is therefore the first to examine the impact of two different model resolutions: a coarse (~ 140 km) and a finer resolution (~ 50 km) on $O_3$ and PM$_{2.5}$ concentrations, and their subsequent impacts on European-scale human health through long-term exposure to $O_3$ and PM$_{2.5}$. We define the sensitivity of health impacts to model resolution by calculating the attributable fraction (AF) of total mortality which is associated with long-term exposure

to $O_3$ and $PM_{2.5}$ for various European countries, based on simulated concentrations at both resolutions, and expressed as a percentage.

The remainder of the paper is organised as follows. Section 2 describes the modelling framework used for both the coarse and finer simulations and the methods used to calculate the AF of mortality associated with $O_3$ and $PM_{2.5}$ for various European countries. Section 3 presents differences in seasonal mean $O_3$ and $PM_{2.5}$ concentrations between the two resolutions. In section 4, we first analyse differences in warm season daily maximum 8-hour running mean (MDA8) $O_3$ concentrations and annual $PM_{2.5}$ concentrations between the two resolutions, then quantify differences in country-level population-weighted MDA8 $O_3$ and annual mean $PM_{2.5}$ concentrations. Secondly, the country-level AF associated with long-term exposure to MDA8 $O_3$ and annual mean $PM_{2.5}$ simulated at both resolutions are presented. The conclusions of this study are then presented in Section 5.

## 2 Methods

### 2.1 Model description and experimental setup

The two chemistry-climate configurations used in this study are based on the Global Atmosphere 3.0 (GA3.0) / Global Land (GL3.0) configuration of the Hadley Centre Global Environmental Model version 3 (HadGEM3, Walters et al., 2011), of the Met Office's Unified Model (MetUM, Brown et al., 2012). The coarse configuration has a horizontal resolution of $1.875° \times 1.25°$ ($\sim 140$ km, Walters et al., 2011) while the finer configuration has a horizontal resolution of $0.44° \times 0.44°$ ($\sim 50$ km, Moufouma-Okia and Jones, 2014) with a domain covering most of Europe.

As this study focuses on health impacts, our analysis is restricted to European land regions. In both configurations, a 63 level hybrid height vertical co-ordinate system is used with 50 levels below 18 km and a surface level at 40 m. Gas phase chemistry is simulated within HadGEM3 by a tropospheric configuration of the United Kingdom Chemistry and Aerosol (UKCA) model (Morgenstern et al., 2009; O'Connor et al., 2014). The chemistry scheme used for both configurations is the UKCA Extended Tropospheric Chemistry (UKCA-ExtTC) scheme (Folberth et al., In prep.) which is an extension to the TropIsop standard chemistry scheme (O'Connor et al., 2014) and includes 89 chemical species. Boundary layer mixing for both configurations is based on Lock et al. (2000) and includes an explicit entrainment parametrisation and non-local mixing in unstable layers. The GA3.0/GL3.0 configuration of HadGEM3 (Walters et al., 2011) also includes an interactive aerosol scheme called CLASSIC (Coupled Large-scale Aerosol Simulator for Studies in Climate; Jones et al., 2001;Bellouin et al., 2011) from which $PM_{2.5}$ concentrations are estimated. CLASSIC simulates ammonium sulphate and nitrate, fossil-fuel organic carbon (FFOC), mineral dust, soot and biomass burning (BB) aerosol interactively. Biogenic secondary organic aerosols are prescribed from a climatology. Sea salt aerosol is diagnosed over ocean only and does not contribute to particulate matter over land.

The model simulations for both these configurations cover a period of one year and 9 months starting from April 2006, from which the first nine months were discarded as spin-up. The coarse configuration uses monthly mean distributions

of sea surface temperature (SST) and sea ice cover (SIC), derived for the present-day (1995-2005) from transient coupled atmosphere-ocean simulations (Jones et al., 2001) of the HadGEM2-ES model (Collins et al., 2011). Using a simple linear re-gridding algorithm, the SST and SIC climatologies developed for the coarse configuration were downscaled to the finer configuration. The coarse configuration was set to produce lateral boundary conditions (LBCs) at six-hourly intervals which were then used to drive the finer configuration.

A consistent set of baseline emissions have been used for both configurations by using the same source data and then re-gridding to the coarse and finer resolutions of the chemistry-climate model. The surface emissions for chemical species were implemented from emission data at 0.5° by 0.5° resolution developed by Lamarque et al. (2010) for the Fifth Coupled Model Inter-comparison Project (CMIP5) report which include reactive gases and aerosols from anthropogenic and biomass burning sources. Both model configurations are driven by decadal mean present-day emissions from Lamarque et al. (2010), representative of the decade centred on 2000. Biogenic emission of isoprene and monoterpenes are calculated interactively following Pacifico et al. (2011) and the biogenic emissions of methanol and acetone are prescribed, taken from Guenther et al. (1995). A full description of other biogenic emissions and the coarse and finer configurations can be found in Neal et al. (2017).

The two configurations are consistent in terms of driving meteorology and emissions as discussed above, however a change in model resolution also requires changes to model's dynamical time-step (from 20 min; coarse resolution to 12 min; finer resolution) as well as some of the parameters in the model parametrisations schemes that are resolution dependent. In this study we assume any such differences to be a model resolution effect. To compare pollutant concentrations simulated at the two resolutions, the coarse model results were re-gridded to the finer resolution via bi-linear interpolation and differences between the two configurations were then calculated at each grid box. For consistency, all figures, tables and values shown in the following sections show differences calculated as coarse minus finer results. All pollutant concentrations used in this study have been extracted at the lowest model level with a mid-point at 20 m. While this level is considered representative of surface or ground-level concentrations, local orographically driven flows or sharp gradients in mixing depths cannot be represented at this vertical resolution (Fiore et al. 2009).

## 2.2 Measurement data

Modelled seasonal mean $O_3$ and $PM_{2.5}$ concentrations for 2007 were evaluated using measurement data from the European Monitoring Evaluation Programme (EMEP) network (ebas.nilu.no). We note that all EMEP stations are classified based on a specific distance away from emission sources so as to be representative of larger areas. For example the minimum distance from large pollution sources such as towns and power plant is ~ 50 km ( Tørseth et al., 2012; EMEP/CCC, 2001). We chose a sub-set of the available EMEP $O_3$ measurement sites with an altitude less than or equal to 200 m above sea level to focus on near-surface comparisons between measurements and simulated $O_3$ concentrations (52 sites – Fig. 1). As there are fewer measurements of $PM_{2.5}$ for 2007, all available EMEP measurement sites were used for $PM_{2.5}$ evaluation (25 sites – Fig. 1). All modelled $O_3$ and $PM_{2.5}$ concentrations shown in this study were taken from the lowest vertical model level which reaches a

height of 40 m. To perform an observation-model comparison, simulated pollutant concentrations were extracted at measurement site locations using bi-linear interpolation.

## 2.3 Health calculations

Annual total mortality estimates associated with long-term exposure to $O_3$ and $PM_{2.5}$ are frequently calculated by estimating the country-level Attributable Fraction (AF) of mortality, based on concentration-response relationships associated with each pollutant, and then multiplying the AF by the baseline mortality rate. Since we are interested in the effects of changing resolution on pollutant concentration, in our analysis, we focus on the absolute values and differences in the AF between the two resolutions, rather than calculating mortality associated with each pollutant, which also depends on underlying baseline mortality rates. This allows us to isolate the effect of model resolution on health impacts. We note that differences in AF will be the same as the differences in mortality between the two resolutions (expressed as a percentage of total mortality), if calculated as described in this section.

Although there is limited evidence available for the long-term health impacts of $O_3$ especially in Europe (The UK Committee on the Medical Effects of Air Pollution (COMEAP) 2015), a number of studies have quantified the adverse health impacts associated with long-term exposure to $O_3$. In this study we apply the Health Risks of Air Pollution in Europe – HRAPIE project recommended coefficient for long-term exposure to $O_3$ (WHO, 2013) to investigate the sensitivity of health calculations to the model resolution used to simulate $O_3$ concentrations. This concentration–response coefficient is derived from the single-pollutant analysis of the American Cancer Society Cancer Prevention Study II (CPS II) cohort study data in 96 metropolitan areas of the US (Jerrett et al., 2009) which has been used by previous studies (e.g. Anenberg et al., 2009; Punger and West, 2013; Thompson et al., 2014; Cohen et al., 2017); but is re-scaled from 1-hour mean to 8-hour mean concentrations using the ratio 0.72, derived from the APHEA-2 project (Gryparis et al., 2004). The value recommended by HRAPIE for the concentration-response coefficient, or $\beta$ value (Eq.1), for the effects of long-term $O_3$ exposure on respiratory mortality is 1.014 (95% Confidence Interval (CI) = 1.005, 1.024) per 10 µg m$^{-3}$ increase in MDA8 $O_3$ during the warm season (April-September) with a threshold of 70 µg m$^{-3}$ (WHO, 2013). For estimating the health impact of long-term exposure to $PM_{2.5}$ on all-cause (excluding external) mortality, HRAPIE (WHO 2013) recommends a relative risk coefficient of 1.062 (95% CI = 1.040, 1.083) per 10 µg m$^{-3}$ increase in annual average concentrations (with no threshold) which is based on a meta-analysis of cohort studies by Hoek et al. (2013)

For MDA8 $O_3$, the risk estimates above, suggested by HRAPIE, are based on data from the American Cancer Society (ACS) cohort (Jerrett et al., 2009) during the warm season re-scaled from 1-hour means to 8-hour means (WHO, 2013). Since MDA8 $O_3$ concentrations in the summer months exceed 70 µg m$^{-3}$ for most areas included in the ACS study, little information exists on the shape of the concentration-response relationship at low levels. For this reason, following HRAPIE suggestions, only MDA8 $O_3$ concentrations exceeding 70 µg m$^{-3}$ and averaged between April and September were used in the present study

to calculate $O_3$-related health impacts. For $PM_{2.5}$-related health impacts we use annual averages with no threshold. As the β values used for $O_3$ and $PM_{2.5}$ are from the ACS cohort, the estimates in this study exclude people younger than 30 years.

For each model resolution, simulated air pollutant concentrations were used to calculate the country-average AF of respiratory or all-cause mortality associated with long-term exposure to $O_3$ and $PM_{2.5}$, respectively. Specifically, the country-average AF is derived from the country-averaged population-weighted pollutant concentration ($x_{country}$) and concentration-response coefficient (β) as shown in Eq. (1) (e.g. Anenberg et al. 2010; Gowers et al. 2014):

$$AF_{country} = 1 - e^{-\beta x_{country}} \qquad (1)$$

The country-averaged population-weighted pollutant concentrations ($x_{country}$) were derived using gridded population data at a resolution of 5 km (GWPv3), obtained from the Socioeconomic Data and Applications Centre (SEDAC, http://sedac.ciesin.columbia.edu/data/set/gpw-v3-population-count-future-estimates/data-download), following Eq. (2).

$$x_{country} = \frac{\sum_{i \,\in country} (p_i \times x_i)}{\sum_{i \,\in country} p_i} \qquad (2)$$

Here, $x_i$ represents the pollutant concentration within each model grid-cell i and $p_i$ represents the total population (aged 30+ years) summed within each model grid-cell. For population-weighted $PM_{2.5}$ concentrations, the simulated $PM_{2.5}$ concentration for each model grid-cell was multiplied by the number of people within the same model grid-cell. This product was then summed for all grid-cells within the country and divided by the total population of the respective country. A similar procedure was used for MDA8 $O_3$ concentrations. However, for populated–weighted MDA8 $O_3$ concentrations, 70 µg m$^{-3}$ was first subtracted from the simulated MDA8 $O_3$ concentration at each grid-cell before multiplying by the population (any resultant negative concentrations were set to zero).

## 3 The impact of model resolution on pollutant concentrations

### 3.1 The impact of model resolution on seasonal mean $O_3$: comparison with observations

Modelled and observed means and, standard deviations (SD), normalised mean bias (NMB) and percentage differences between the two resolutions for all four seasons at the 52 EMEP site locations are shown in Table 1. Similarly modelled means, SD and percentage differences between the two resolutions are also shown for all model cells within the European domain (discussed in Section 3.2). Compared to measurements, mean values simulated by the chemistry-climate model across the 52 station locations are lower in winter (DJF) and higher in summer (JJA) and autumn (SON) with NMB values up to -19%, 27% and 19%, respectively. In spring (MAM), simulated mean $O_3$ concentrations at the finer resolution are closest to observations (NMB = ~ -4 %), whilst in all other three seasons the simulated values at the coarse resolution are in closer agreement with observations (NMB = ~ -8%, ~24% and ~ 5%, respectively).

For all seasons, the SD of seasonal mean $O_3$ concentrations, simulated at the two resolutions are more similar to each other than to observations. However, the SD across all 52 sites, simulated at the coarse resolution is higher than that simulated at the finer resolution.

Modelled versus observed seasonal mean $O_3$ concentrations for each of the 52 EMEP station locations are shown in Fig. 2, with arrow lengths indicating the change in concentrations when simulated at the coarse versus finer resolutions. For both resolutions, higher $O_3$ concentrations are simulated during summer compared to observations as noted above (between 50 to 150 µg m$^{-3}$; Fig. 2). In winter, simulated $O_3$ concentrations are lower compared to measurements (< 30 µg m$^{-3}$), and are most similar to observations in spring and autumn in accordance with lower NMB (Table 1).

The magnitude of the differences in simulated $O_3$ concentrations between the two resolutions varies seasonally, with the smallest (coarse-finer) differences in summer (green arrows – Fig. 3; -3 % ;Table 1) and the largest difference in spring, as noted above (16 % ;Table 1). Similar differences in July mean $O_3$ concentrations between a 150 km and a 40 km resolution were also found by Stock et al. (2014). Over the majority of the stations, during winter and spring, $O_3$ concentrations simulated at the finer resolution are lower than concentrations simulated at the coarse resolution (downward arrows; Fig. 2, positive difference; Table 1). In contrast during summer and autumn, $O_3$ concentrations are higher when simulated at the finer resolution (upward arrows; Fig. 2, negative difference; Table 1). These results are analysed further at the seasonal level in Fig. S1 of the Supplement to this article (Supplement S1; Fig. S1).

## 3.2 The impact of model resolution on seasonal mean $O_3$: spatial differences

This section extends our investigation to examine the impact of model grid resolution on the spatial distribution of $O_3$ over the whole of Europe. The seasonal variation in $O_3$ concentrations simulated at the finer resolution across Europe shows the same features as at the 52 site locations (section 3.1), with highest values in spring and summer (> 50 µg m$^{-3}$ and up to 120 µg m$^{-3}$; Fig. 3b and 3c, respectively) and lowest values in autumn and winter (<55 µg m$^{-3}$; Fig. 3a and 3d). In all seasons, except winter, there is a clear latitudinal gradient with higher $O_3$ concentrations in southern compared to northern Europe. In winter (Fig. 3a), very low $O_3$ concentrations are simulated across much of Europe (~30 µg m$^{-3}$).

For most of Europe, in winter and spring, mean $O_3$ concentrations are generally higher when simulated at the coarse compared to the finer resolution (Fig. 3e and 3f, 10% and 6% respectively; Table 1), in agreement with the findings for the sub-set of 52 locations. However parts of northern Scandinavia and the UK, and parts of south-eastern Europe have lower $O_3$ concentrations simulated at the coarse resolution in these two seasons. In summer and autumn, $O_3$ concentrations are slightly lower when simulated at the  coarse compared to the finer resolution (-1% and -4% respectively –Table 1) as found for the sub-set of locations, except in areas of easternmost Europe (especially in autumn) and parts of Spain and Italy (Fig. 3g and 3h). The greatest positive differences in simulated $O_3$ concentrations, i.e. higher values at the coarse resolution, are found in winter, especially in the far south of Europe in Spain (~ 20 µg m$^{-3}$; Fig. 3e). Some of these positive differences are clear around the coastal regions which is likely due to differences in the land/sea mask at the two resolutions, which leads to less deposition over oceanic grid-cells at the coarse resolution and higher simulated $O_3$ concentrations compared to the same locations that

are designated as land at the finer scale (Coleman et al., 2010). In addition, large positive differences in simulated $O_3$ concentrations between the two resolutions occur over the Alps, whereby simulated $O_3$ concentrations are higher at the finer scale (Fig. 3e and 3h). This is most likely due to the differences in orography at the two resolutions with higher elevations at the finer scale leading to higher $O_3$ concentrations.

Differences in simulated seasonal mean $NO_2$ concentrations at the two resolutions show similar, but less extensive differences and generally inverse patterns as for $O_3$ concentrations, with some negative differences, i.e. lower $NO_2$ values in winter and spring (Fig. 3i and 3j), when simulated at the coarse compared to the finer resolution. In contrast, in summer and autumn, $NO_2$ concentrations are higher in some regions when simulated at the coarse compared to the finer resolution (e.g. Italy; Fig. 3k and 3l). An inverse relationship i.e. a positive difference in $O_3$ concentrations and a negative difference in $NO_2$

concentrations is most prominent for locations in Spain (all year around) and Italy (winter and spring) and parts of the Benelux region (southern UK and Netherlands; all year around). This inverse relationship is driven by lower $NO_x$ concentrations at the coarse resolution which lead to less $O_3$ titration by NO compared to the finer resolution (Fig. 3i). This in turn results in higher simulated seasonal mean $O_3$ concentrations at the coarse resolution compared to the finer resolution (Fig. 3e).

The planetary boundary layer (PBL) height is a key meteorological variable that affects the vertical transport of

pollutants from the surface into the free troposphere from where they can then undergo strong horizontal transport. Thus we have also investigated the impact of changing model resolution on PBL height and how this impacts $O_3$ and $NO_2$ concentrations. Spatial differences in PBL height between the two resolutions are shown in Section S2, Fig. S2 of the Supplement to this article. In all seasons, over most of western and central Europe and especially in summer, the PBL height is generally lower when simulated at the coarse resolution (negative differences up to 275m; Fig. S2c). In winter and spring

(Fig. S2a and S2b), this lower height corresponds to generally higher $O_3$ concentrations but also lower $NO_2$ concentrations simulated at the coarse resolution, over the same region and vice versa in summer and autumn (Fig. S2c and S2d). If a deeper PBL is the main driver of pollutant trapping producing higher $O_3$ levels, then we would also expect $NO_2$ concentrations to be higher with a lower PBL height at the finer resolution, but their frequent inverse relationship suggest a stronger role for chemistry rather than PBL effects. However, these chemical and physical processes cannot be clearly separated.

In summary, we find a seasonal variation in simulated $O_3$ differences between the two resolutions. Simulated $O_3$ concentrations at the coarse resolution are higher in winter and spring and lower in summer and autumn compared to the finer resolution. We also find that in a number of locations, $NO_2$ concentrations are lower at the coarse compared to the finer resolution and correspond to higher $O_3$ concentrations at the coarse resolution as a result of reduced titration with lower $NO_x$ levels. Orography also plays an important role in some coastal locations, leading to an overestimation of $O_3$ concentrations.

The PBL height differs between the two resolutions especially during summer, with the finer resolution resulting in a deeper boundary layer. However, it is not possible to separate chemistry and mixing effects on simulated $O_3$ concentrations.

**3.3 The impact of model resolution on seasonal mean PM$_{2.5}$ – comparison with observations**

Simulated seasonal mean PM$_{2.5}$ concentrations are compared to available EMEP observations at 25 sites (Table 2). Mean values for the observations are fairly similar across all seasons, with values in summer and autumn being slightly lower. PM$_{2.5}$ concentrations simulated at both the coarse and finer resolutions are lower in winter and higher in summer compared to measurements. In addition, mean PM$_{2.5}$ concentrations simulated at the finer resolution are higher than those simulated at the coarse resolution except in summer. The coarse resolution simulates PM$_{2.5}$ levels with the smallest bias during spring (NMB = -0.2%). In contrast, PM$_{2.5}$ concentrations simulated at the finer resolution during spring have a large positive bias (NMB = 31%). Similarly in autumn NMB values are larger for PM$_{2.5}$ concentrations simulated at the finer resolution. We find that the largest bias for both resolutions occurs in summer with the coarse resolution resulting in a NMB of 70%. Using, a similar finer configuration, Neal et al. (2017) found a year-round small positive bias in simulated PM$_{2.5}$ concentrations averaged over a five year period (2001-2005) at two UK locations. The SD of PM$_{2.5}$ concentrations across the 25 sites is fairly similar between model results and measurements except in winter, when simulated SD values are lower at both resolutions compared to measurements and in autumn, when the SD at the finer resolution is higher compared to measurements.

Modelled versus measured PM$_{2.5}$ concentrations across the 25 individual EMEP stations highlight the low simulated PM$_{2.5}$ concentrations in winter (Section S2, Fig. S3 of the Supplement to this article). Large variations in PM$_{2.5}$ levels between the two resolutions are prominent in spring (-31%; Table 2). Smaller PM$_{2.5}$ concentrations simulated at the coarse resolution in winter, spring and autumn are apparent (upward arrows; Fig. S3, negative differences; Table 2).

**3.4 Impact of model resolution on seasonal mean PM$_{2.5}$: spatial differences**

Spatial distributions of PM$_{2.5}$ concentrations, simulated at the finer resolution as well as differences between the two resolutions over the whole European domain are illustrated in Fig. 4. Over the whole domain, PM$_{2.5}$ concentrations simulated at the finer resolution are lowest in winter (Fig. 4a) and highest in spring (Fig. 4b). As for O$_3$, there is clear latitudinal gradient with higher PM$_{2.5}$ levels in southern Europe in all seasons. Differences in seasonal mean PM$_{2.5}$ concentrations, between the coarse and fine resolutions, vary seasonally across the European domain with the smallest differences occurring during winter ($\pm$ 3 µg m$^{-3}$; Fig. 4e, -8% ; Table 2) in agreement with the findings for the 25 EMEP stations described above (section 3.3). This suggests that at low PM$_{2.5}$ concentrations (~ 8 µg m$^{-3}$) in winter, model results do not differ greatly when increasing the model resolution from 150 km to 50 km. In spring, PM$_{2.5}$ concentrations simulated at the coarse are lower than at the finer resolution over large parts of central and western Europe but are slightly higher in easternmost parts of Europe (negative differences ~-10 µg m$^{-3}$ Fig. 4f; -6% Table 2), as found at the 25 EMEP station locations. The opposite result occurs in summer with generally higher PM$_{2.5}$ concentrations simulated at the coarser resolution (positive differences ~ 10 µg m$^{-3}$ Fig. 4g; 29% Table 2). In autumn, the differences in PM$_{2.5}$ concentrations at the two resolutions exhibit a marked east-west contrast, with lower values at the coarse resolution in western Europe (where the EMEP stations are generally located; Fig. 1) and higher values at the coarse resolution in eastern Europe (Fig. 4h). While PM$_{2.5}$ concentrations at the 25 EMEP site locations are on average lower when

simulated at the coarse resolution (-23%), over all grid-cells, PM$_{2.5}$ concentrations are higher at the coarse resolution (8%).
This highlights issues with representivity of the EMEP network across Europe, with much fewer EMEP measurement stations for PM$_{2.5}$ in eastern Europe.

The seasonality in PM$_{2.5}$ differences, brought about by a change in model horizontal resolution, can be partly explained by differences in PBL height between the two resolutions, as outlined in section 3.2. In particular, the deeper boundary layer in summer simulated at the finer resolution may lead to greater vertical lofting from the surface, producing lower PM$_{2.5}$ levels compared to that simulated at the coarse resolution. In addition, differences in simulated precipitation (especially smaller-scale convective precipitation) between the two resolutions may be important, through its influence as the dominant mechanism in UKCA for removal of aerosols through wet deposition (O'Connor et al., 2014). Spatial patterns of convective precipitation differences between the two resolutions are shown in Section S2, Fig. S4 of the Supplement to this article. In winter and spring, convective rainfall is higher at the coarse compared to the fine resolution (Fig. S4a and S4b). Thus removal of PM$_{2.5}$ through wet deposition is greater, producing lower PM$_{2.5}$ concentrations at the coarser resolution (Fig. 4e and 4f). The opposite holds in summer and autumn as the convective rainfall is lower at the coarse compared to the finer resolution (Fig. S4c and S4d) therefore resulting in higher PM$_{2.5}$ concentrations simulated at the coarse resolution (Fig. 4g and 4h).

Overall, we also find a large seasonal variation in simulated PM$_{2.5}$ concentrations between the two resolutions, with typically lower levels simulated in winter and spring at the coarse compared to the finer resolution and the opposite result in summer and autumn. Hence, the seasonality of differences in simulated PM$_{2.5}$ concentrations between the two model resolutions is generally the inverse of that found for O$_3$ in section 3.3. We find that these seasonal differences can be largely explained by meteorological effects: PBL height differences, especially in summer, and by differences in convective rainfall between the two resolutions.

## 4 Sensitivity of health impact estimates to model resolution

We now examine how the differences in O$_3$ and PM$_{2.5}$ concentrations simulated at the two resolutions, influence health impact estimations across Europe at the country level. For this analysis we use warm season daily maximum 8-hour running mean (MDA8) O$_3$ (above 70 µg m$^{-3}$) and annual-average PM$_{2.5}$ concentrations. To estimate health impacts, air pollution concentrations (with an averaging period consistent with that used in epidemiological studies) are combined with population estimates and concentration-response coefficients (Section 2.3).

### 4.1 Warm season MDA8 O$_3$ and annual-average PM$_{2.5}$ concentrations

Statistics for warm season MDA8 O$_3$ and annual PM$_{2.5}$ concentrations compared between EMEP measurements and model results at the two resolutions are provided in Table 3. Mean simulated MDA8 O$_3$ levels in the warm season at the 52 EMEP locations for both resolutions, are higher compared to observations (NMB = 11% and 9 %; Table 3), in agreement with our

findings for summer and autumn mean $O_3$ levels (c.f., Table 3, Table 1). The SD is also higher for both resolutions compared to observations. However, in contrast with summer mean $O_3$ levels, mean simulated MDA8 $O_3$ concentrations are 0.8% higher at the coarse compared to the finer resolution at the 52 EMEP site locations (Table 3). Simulated annual mean $PM_{2.5}$ concentrations are also higher compared to observations at the 25 locations (NMB =~10-20%; Table 3) with concentrations being 8.7% lower at the coarse compared to the finer resolution. This represents the net effect of seasonality in NMB shown in Table 2.

Differences in warm season MDA8 $O_3$ and annual mean $PM_{2.5}$ concentrations, simulated at the coarse and finer resolution, are shown in Fig. 5. The spatial distribution of differences in warm season MDA8 $O_3$ between the two resolutions (Fig. 5a) is most similar to the distribution of differences in summer mean $O_3$ concentrations (Fig. 3g). Differences in MDA8 $O_3$ concentrations range from ~ -7 µg m$^{-3}$ in Northeast Europe to ~ +20 µg m$^{-3}$ in Southern Europe, UK and Ireland (Fig. 5a). We note that if a different time-averaging period was chosen e.g., annual as opposed to warm season, the spatial patterns of MDA8 $O_3$ differences would alter considerably due to the seasonal variation displayed in Figure 3.

The spatial distribution of differences in annual mean $PM_{2.5}$ concentrations between the two resolutions (Fig. 5b) are most similar to the spatial distribution of differences in spring and especially autumn mean $PM_{2.5}$ concentrations notably with an east-west gradient (Fig. 5). Differences in $PM_{2.5}$ concentrations between the two resolutions range from ~ -8 µg m$^{-3}$ in the southwestern part of Europe and Cyprus to ~ +4 µg m$^{-3}$ in north and eastern Europe (Fig. 5b).

## 4.2 Effect of applying population-weighting to MDA8 $O_3$ and annual $PM_{2.5}$ concentrations

The warm season MDA8 $O_3$ concentrations and annual mean $PM_{2.5}$ concentrations, simulated at both resolutions, were weighted by population totals for each country to produce country average population-weighted concentrations (Section 2.3). Figure 6a shows the impact of the two resolutions on country-average warm season average MDA8 $O_3$ and the corresponding population-weighted MDA8 $O_3$ concentrations. Similarly differences in annual mean $PM_{2.5}$ concentrations between the two resolutions for non-population-weighted and population-weighted concentrations are shown in Fig. 6b.

Population-weighting of pollutant concentrations has different impacts across the European countries (Fig. 6a and 6b). In many countries, differences in population-weighted pollutant concentrations between the two resolutions are enhanced (i.e. larger positive or more negative differences) relative to non-population-weighted pollutant concentrations. However, in some countries population-weighting may reduce the positive or negative difference between the two resolutions. We examine several cases below.

For warm season MDA8 $O_3$ concentrations, the largest negative differences, implying lower MDA8 $O_3$ levels using coarse compared to the finer resolution results, occur in eastern Europe (Fig. 5a). Hence, the largest negative differences in non-population-weighted and population-weighted MDA8 $O_3$ concentrations are found in eastern European countries (Fig. 6a). The difference between the two resolutions is greatest when population-weighting is applied. This is generally due to slightly lower population-weighted MDA8 $O_3$ concentrations compared to MDA8 $O_3$ concentrations derived from the coarse resolution results (Section S3, Fig. S5a of the Supplement to this article).

In the Netherlands warm season non-population-weighted MDA8 $O_3$ is also lower when derived from coarse compared to finer resolution results (negative difference; Fig. 5a, 6a). However population-weighted MDA8 $O_3$ concentrations are higher when derived from the coarse resolution results (Fig. 6a). This is caused by lower MDA8 $O_3$ concentrations simulated at the finer resolution when applying population-weighting (Fig. S5a). This suggests that in populated regions, MDA8 $O_3$ concentrations simulated at the finer resolution are lower which might be linked to higher $NO_2$ concentrations.

Warm season MDA8 $O_3$ show the largest positive differences, with higher values simulated at the coarse resolution, for southern Europe and the UK/Ireland (Fig 5a). Thus, the largest positive differences for non-population-weighted and population-weighted MDA8 $O_3$ concentrations occurs in south European countries (Fig. 6a). Population–weighed MDA8 $O_3$ concentrations in Portugal are higher compared to MDA8 $O_3$ concentrations at the coarse but lower at the finer resolution (Fig. S5a). This suggests that, at the coarse resolution, areas with high levels of $O_3$ are co-located with high population densities whilst at the finer resolution areas with lower levels of $O_3$ are co-located with high population densities.

Annual-average $PM_{2.5}$ concentrations show the largest negative differences, with higher values simulated at the finer resolution, in parts of western Europe (Fig. 5b). Hence, the largest negative non-population-weighted and population-weighted annual mean $PM_{2.5}$ concentrations are found for Cyprus, Italy and Spain (Fig. 6b). Conversely, higher annual-average $PM_{2.5}$ levels are simulated at the coarse resolution in eastern and northern Europe (Fig. 5b), hence larger positive non-population-weighted and population-weighted annual mean $PM_{2.5}$ concentrations occur for countries in eastern Europe and northern Europe (Fig. 6b).

In Cyprus, population-weighted annual mean $PM_{2.5}$ concentrations simulated at the fine resolution are higher compared to concentrations with no population-weighting, due to denser populations being co-located with areas of higher $PM_{2.5}$ levels (Fig. S5b). In Croatia, population-weighted annual mean $PM_{2.5}$ concentrations simulated at the coarse resolution are greater than $PM_{2.5}$ concentrations with no population-weighted, again due to denser populations in regions of high concentrations but in this case when simulated at the coarse resolution (Fig. S5b). In a few countries (e.g. Switzerland), differences in population-weighted annual mean $PM_{2.5}$ concentrations between the two resolutions have an opposite sign to differences between concentration with no population-weighting (Fig. 6b). This indicates that annual mean $PM_{2.5}$ concentrations simulated at the finer resolution are high in densely populated regions but are low in these same regions at the coarse resolution.

It would be insightful to examine these population-weighted results in relation to model-observation biases in densely populated areas. However, as outlined in Section 2.2, the available sites in the EMEP database are urban background stations which are required to be representative of a wide are and away from industrial areas (EMEP/CCC,2001). Nonetheless we do note that in southern Europe, simulated summer mean MDA8 $O_3$ concentrations at the finer resolution are closer to observations than concentrations simulated at the coarse resolution. We find no consistent result for model biases in simulated annual mean $PM_{2.5}$ concentrations with respect to observations for the two model resolutions.

## 4.3 Attributable fraction of mortality associated with long-term exposure to O₃

The Attributable Fraction (AF) associated with long-term exposure to MDA8 $O_3$, expressed as a percentage of total respiratory mortality and simulated at both resolutions, was calculated for each country (Fig. 7a), using the population-weighted warm season MDA8 $O_3$ concentrations (Fig. 6a) as discussed in Section 2.3. For both resolutions, the estimated AF is shown for each country, with the 95% confidence interval (95% C.I.) representing uncertainties associated only with the concentration-response coefficient (β) used (shown in grey). For all the countries considered, irrespective of the model resolution used, the AF of total respiratory mortality ranges from 1% (95% C.I. 0% - 2%) in Finland to 11 % (95% C.I. 4% - 18%) in Cyprus (Fig. 7a).

Differences in AF between the countries are solely attributed to differences in population-weighted MDA8 $O_3$ concentrations. Thus, countries with the highest population-weighted concentrations also have the highest AF. Similarly countries with the highest differences in population-weighted MDA8 $O_3$ concentrations between the two resolutions also have the largest differences in AF between the coarse and finer resolution. If the AF was calculated for each model grid-cell rather than at the country level, the differences in AF for the two pollutants would have identical spatial distributions to the differences in warm season MDA8 $O_3$ and annual-mean $PM_{2.5}$ concentrations depicted in Fig. 5, as the AF is only dependent on the pollutant concentration and β (which is constant across all countries).

The differences in country level AF associated with long-term exposure to warm season MDA8 $O_3$, simulated at the two resolutions, are shown in Fig. 7b. These values highlight the sensitivity of respiratory mortality attributable to long-term exposure to $O_3$ to a change in model resolution. For most of northern and eastern Europe, the AF at the coarse resolution is lower than that at the finer resolution (negative differences; Fig. 7b) as for differences in population-weighted warm season MDA8 $O_3$ concentrations in the same countries (Fig. 6a). In contrast, the AF at the coarse resolution is higher than that at the finer resolution for countries in southern Europe (positive differences, Fig. 7b). Differences in AF range from -0.9% (95% C.I. -0.3% to -1.5%) in Poland to +2.6% (95% C.I. 1.0% to 4.1%) in Portugal (Fig. 7b) which directly correspond to the countries having the lowest and highest difference in population-weighted MDA8 $O_3$ concentration respectively (Fig. 6a; Note, although the differences in AF between the two resolution appear low, these are percentages of total respiratory mortality). In Poland and Portugal the estimated AF at the finer resolution is 1.4 times and 0.7 times that estimated at the coarse resolution. For approximately half of the European countries, the AF is higher for the coarse resolution compared to the finer resolution and vice versa. When considering the uncertainty associated with the concentration-response coefficient used, the sign of the difference of AF between the two model resolutions is unaltered (Fig. 7b). Over the majority of the countries, the AF attributable to long-term exposure to MDA8 $O_3$ by the coarse resolution fall within the range of uncertainty as calculated by the finer resolution (Fig. 7a). However, over Finland and Ireland, the coarse mean estimates fall outside the uncertainty range estimates using the finer resolution (Fig. 7a).

For U.S. averaged mortality estimates, Punger and West (2013) show that mortality estimates related to warm season long-term $O_3$ exposure, calculated using the $O_3$ concentrations at 36 km, were higher (by 12%) than estimates calculated at the

12 km resolution. Resolution was also found to play and important role in determining health benefits associated with differences in $O_3$ between 2005 and 2014 in the U.S. (Thompson et al. 2014). In particular, in urban areas, Thompson et al.

(2014) estimate that the benefits calculated using coarse resolution results were on average two times greater than estimates calculated using the finer scale results. Both the studies mentioned are conducted in the U.S. and use a different concentration response coefficient and thus a definitive comparison between these studies and our estimates over Europe is not possible.

Since, seasonal differences in simulated $O_3$ with resolution are considerable, the AF associated with long-term

exposure to $O_3$ was also calculated based on annual-mean (as opposed to summer-mean) $O_3$ concentrations based on recommendations by Turner et al. (2015). Turner et al (2015) suggest a higher concentration response coefficient of 1.06 (95% CI: 1.04-1.08) per 10 µg m$^{-3}$ and a slight lower MDA8 $O_3$ threshold of 53.4 µg m$^{-3}$ compared to values used in our study for summer-mean MDA8 $O_3$. Using the values from Turner et al. (2015) the differences in AF are found to be of the same sign for the majority of the countries and the rankings across countries are largely similar. This similarity occurs because the

difference in annual-mean MDA8 $O_3$ concentrations between the two resolutions shows generally similar spatial patterns to the differences in warm season MDA8 $O_3$ concentrations (not shown). However the ranges when using annual-mean $O_3$ concentrations and recommendations from Turner et al. (2015) are larger: -2.3% to +12.0%, compared to AF ranges given above for MDA8 $O_3$. From further sensitivity analyses it is found that these greater AF ranges can be attributed to the use of a higher concentration-response coefficient (by a factor of approximately 4) rather than differences in annual-mean compared

to summer-mean concentrations.

**4.4 Attributable Fraction associated with long-term exposure to PM$_{2.5}$**

The fraction of all-cause (excluding external) mortality attributable to long-term exposure to PM$_{2.5}$, is shown as percentages for each country in Fig. 8a. The AF for all countries, irrespective of the resolution used, ranges from 2% (95% C.I. 1% - 3%) in Iceland to 15% (95% C.I. 10% - 19%) in Cyprus (Fig. 8a). Differences in AF between the two resolutions are shown in Fig.

8b. Since the variability in AF differences across the countries is caused by variability in population-weighted annual mean PM2.5 differences, Cyprus and countries in parts of western Europe have the largest negative difference in percentage AF (Fig. 8b). In contrast, countries in eastern and northern Europe have the largest positive difference in percentage AF (Fig. 8b). These differences range from -4.7% (95% C.I. -6.1% to -3.2%) in Cyprus to 2.8% (95% C.I. 1.9% to 3.7%) in Croatia. For Cyprus and Croatia, using the finer resolution results in an estimated AF that is 1.5 and 0.7 times that estimated using the

coarse resolution. Over most countries, annual mean population-weighted PM$_{2.5}$ concentrations are higher (positive difference; Fig. 6b) for the coarse compared to the finer resolution, thus resulting in a higher AF when using the coarse resolution results. Note, similar to $O_3$, the uncertainty associated with the concentration-response coefficient for PM$_{2.5}$ does not alter the sign of the difference of AF between the two model resolutions (Fig. 8b). For a number of countries, the mean AF attributable to long-term exposure to PM$_{2.5}$ using the coarse resolution falls outside the uncertainty range of the finer estimates in particular over

Iceland and Ireland (Fig. 8a).

We also examine the impact of using a low-concentration threshold. We apply a threshold of 5.8 µg m$^{-3}$ (suggested by Burnett et al. (2014) which is derived from Lim et al. (2012)) to annual mean PM$_{2.5}$ concentrations. Differences in AF estimates associated with long-term exposure to population-weighted PM$_{2.5}$ concentrations range from -4.8% to +2.1% (as compared to -4.7% to +2.8% above when no threshold is applied). The spatial distribution of these estimates remains unchanged and only slight changes in country rankings occur. Hence, the impact of applying a low concentrations threshold in this study for Europe is small.

Our results are consistent with other studies, but not all, that examine the impact of model resolution on health estimates associated with long-term exposure to PM$_{2.5}$. Using concentrations simulated at the 36 km resolution, Punger and West (2013) find that the U.S. national health estimate is higher (11%) than the estimate at 12 km resolution. Li et al. (2015) also show that averaged over the US, a coarse grid resolution (~ 200 km) results in a health estimate that is lower (8%) than the estimated based on the fine scale model results (~ 50 km), in contrast to our findings averaged across Europe. In contrast, Thompson et al. (2014) find that health benefits associated with changes in PM$_{2.5}$ concentrations between 2005 and 2014 in the U.S., were not sensitive to resolution. Both Punger and West (2013) and Li et al. (2015) find that differences in PM$_{2.5}$ are mainly attributable to primary anthropogenic PM, while Thompson et al. (2014) attribute the greatest differences (between 36 km and 4 km resolutions) to secondary PM. However, in our study no substantial differences in PM$_{2.5}$ components between the two resolutions were found. All the mentioned studies are conducted in the U.S. and hence definitive comparisons cannot be made with our results for Europe

In summary, our results suggest that differences in AF health estimates between coarse and finer resolutions vary across the different European countries with clear differences between southern and eastern Europe for exposure to warm season MDA8 O$_3$ and west-east differences for exposure to annual-average PM$_{2.5}$ due to the dependence of AF on populated weighted MDA8 O$_3$ and annual PM$_{2.5}$ concentrations. For differences in AF attributable to long-term exposure to summer mean MDA8 O$_3$ and annual mean PM$_{2.5}$ concentrations, the uncertainty associated with the concentration-response coefficient used does not alter the sign of the difference of AF between the two model resolutions (Fig. 7b and 8b). The uncertainty ranges for the PM$_{2.5}$ –related estimates show a greater variability between the two resolutions for more countries compared to MDA8 O$_3$-related AF estimates. Using the concentration-response coefficient in Jerrett et al. (2009), Thompson et al. (2014) find that the avoided mortalities due to difference in ozone concentrations between 2005 and 2014 at a 36 km model resolution are within the 95% uncertainty range associated with the concentration-response coefficient used compared to estimates at a resolution of 12 km and 4 km. These authors also find avoided mortalities associated with long-term effects of PM$_{2.5}$ exposure at 36 km to fall within estimates at the 12 km and 4 km resolution for three different concentration-response coefficients. Thus our results are in agreement for summer mean O$_3$ but less for annual mean PM$_{2.5}$.

**5 Conclusions**

Chemistry-climate model simulations were performed at two resolutions: a coarse resolution (~ 140 km) and a finer resolution (~ 50 km) over Europe to quantify the impact of horizontal model resolution on simulated $O_3$ and $PM_{2.5}$ concentrations by
season; and on the associated Attributable Fraction (AF) of mortality due to long-term exposure to these two pollutants. Simulated $O_3$ concentrations are lower in winter and higher in summer and autumn compared to measurements at both model resolutions. Results show a seasonal influence in the mean $O_3$ differences between the two resolutions. Simulated $O_3$ concentrations averaged across Europe at the coarse resolution are higher in winter and spring (10% and 6%, respectively), and lower in summer and autumn (-1% and -4%, respectively) compared to the finer resolution. In contrast during winter and
spring, $NO_2$ concentrations are lower in some areas at the coarse compared to the finer configuration, whilst in summer and autumn, there are more locations where $NO_2$ concentrations are higher at the coarse resolution. The lower $O_3$ concentrations simulated at the finer compared to the coarse resolution can be partly explained by these higher $NO_2$ levels that enhance titration of $O_3$ at this finer resolution. The PBL height also differs between the two resolutions and may also account for differences in $O_3$ concentrations; however, it is not possible to clearly separate the effects of chemistry and mixing on simulated
$O_3$.

Differences in $PM_{2.5}$ concentrations simulated at the two resolutions also vary seasonally. Modelled $PM_{2.5}$ concentrations are lower in winter and higher in summer compared to measurements at both resolutions. Simulated seasonal mean $PM_{2.5}$ concentrations averaged across Europe during winter and spring are lower at the coarse compared to the finer resolution (-8% and -6%, respectively) but higher in summer and autumn (29% and 8%, respectively). This seasonality in
Europe-average differences in $PM_{2.5}$ concentrations is opposite to that found for differences in $O_3$ concentrations between the two resolutions. Differences in $PM_{2.5}$ concentrations simulated at the two resolutions are also influenced by PBL height, especially in summer when a deeper boundary layer at the finer resolution leads to greater lofting and lower $PM_{2.5}$ concentrations. Furthermore, in all seasons, the differences in $PM_{2.5}$ levels between the two resolutions are closely related to differences in the convective rainfall rate. In winter and spring, the convective rainfall at the coarse resolution is higher than
that at the finer resolution thus resulting in lower $PM_{2.5}$ concentrations. The opposite result holds in summer and autumn. Results show that differences in warm season mean MDA8 $O_3$ concentrations between the two resolutions are similar to summer mean differences in simulated $O_3$ concentrations, with spatial patterns of differences reveal clear and important contrasts. Warm season MDA8 $O_3$ levels are higher in most of southern Europe as well as the UK and Ireland, but lower in other areas of northern as well as eastern Europe when simulated at the coarse resolution compared to the finer resolution. On
the other hand, annual average $PM_{2.5}$ concentrations are higher across most of northern and eastern Europe but lower over parts of southwest Europe at the coarse compared to the finer resolution.

Weighting the pollutant concentrations at both resolutions with the population within each country, results in some added differences between concentrations at the two resolutions which also vary across the countries. In many countries, weighting by population enhances either positive or negative differences in warm season MDA8 $O_3$ or annual mean $PM_{2.5}$

concentrations between the two resolution, which suggests that high levels of pollutant concentrations coincide with high population density at one resolution but low pollutant concentrations are co-located with high population density at the other resolution. Population-weighting pollutant concentrations also reduces differences between coarse and finer resolution results in some countries.

The AF of respiratory mortality associated with long-term exposure to warm season MDA8 $O_3$ and annual mean
$PM_{2.5}$ is also sensitive to resolution as is it is solely driven by the simulated population-weighted pollutant concentrations. For the AF associated with long-term exposure to $O_3$, countries in northern as well as eastern Europe have lower AF values at the coarse compared to the finer resolution whilst the opposite result occurs for other countries in southern Europe and Ireland. For the AF associated with long-term exposure to $PM_{2.5}$, a few countries in southwestern Europe and Cyprus have lower AF values for $PM_{2.5}$ concentrations simulated at the coarse resolution whilst more countries especially in eastern and northern
Europe show a higher AF when using $PM_{2.5}$ concentrations simulated at the coarse resolution.

Overall, differences in the country-average AF associated with long term exposure to MDA8 $O_3$ range between -0.9 % and +2.6 % while differences in the AF associated with long-term exposure to annual mean $PM_{2.5}$ range from -4.7% to +2.8 % of the total baseline mortality. This result emphasizes the importance of model horizontal resolution when conducting country specific health impact studies. We also find that the impacts of a 95% C.I. in concentration-response coefficient is
smaller than the impact of the model horizontal resolution. In addition, these ranges in AF associated with long-term exposure to annual mean $PM_{2.5}$ were largely unaltered with the application of a low-concentration threshold for $PM_{2.5}$.

Our calculation for $O_3$ health impacts only considers warm-season MDA8 $O_3$ impacts however these may differ to annual MDA8 $O_3$ impacts because of seasonal differences in simulated $O_3$ with resolution highlighted in this study. When using annual-mean MDA8 $O_3$ concentrations alongside a recommended concentration-response coefficient and threshold
suggested by Turner et al. (2015) the difference in AF between the two resolutions is considerably larger than our estimates using summer-mean MDA8 $O_3$ concentrations. This is driven by the higher concentration-response coefficient (by a factor of approximately 4) quoted in Turner et al. (2015) compared to that suggested by HRAPIE for summer mean MDA8 $O_3$ concentrations (WHO, 2013). In addition, for our study we apply the same concentration-response coefficient to all populations and assumed that for $PM_{2.5}$-related health impacts, all $PM_{2.5}$ components have the same impact on mortality.

The pollutant concentrations used in this study have been extracted at the lowest model level with a mid-point at 20 m. The sensitivity of our simulated pollutant concentrations to vertical model resolution has not been examined. Future research focusing on the sensitivity of AF changes to different averaging periods or seasons would be beneficial. In addition, the use of concentration-response coefficients that are derived from European cohort data would be useful, although such data are limited. Nonetheless this study provides one of the first insights as to how air pollution related health impacts over Europe are
influenced by the model resolution used to simulate pollutant concentrations.

**Acknowledgements**

Sara Fenech's PhD was funded by Public Health England. The development of the United Kingdom Chemistry and Aerosol (UKCA) model and Fiona M. O'Connor are supported by the Joint UK BEIS/Defra Met Office Hadley Centre Climate Programme (GA01101).

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

**Table 1: Statistical results comparing seasonal mean O$_3$ concentrations simulated at the coarse and finer resolutions to observations from 52 stations within the EMEP network in 2007. Statistical results for all model grid-cells of both resolutions are also shown. Percentage differences between the two model resolutions are calculated as (O$_3$ $_{coarse\ resolution}$ −O$_3$ $_{finer\ resolution}$)/(O$_3$ $_{coarse\ resolution}$).**

| Season | | Obs. | Model 52 sites | | Model all grid-cells | |
|---|---|---|---|---|---|---|
| | | | 140 km | 50 km | 140 km | 50 km |
| **DJF** | Mean (µg m$^{-3}$) | 52.8 | 48.5 | 42.6 | 35.1 | 31.7 |
| | **Difference in model mean (%)** | | 12.2 | | 9.7 | |
| | NMB (%) | | -8.1 | -19.2 | | |
| | SD (µg m$^{-3}$) | 11.0 | 17.0 | 16.0 | 17.3 | 16.5 |
| **MAM** | Mean (µg m$^{-3}$) | 70.4 | 80.7 | 67.9 | 75.7 | 71.5 |
| | **Difference in model mean (%)** | | 15.9 | | 5.5 | |
| | NMB (%) | | 14.6 | -3.6 | | |
| | SD (µg m$^{-3}$) | 8.9 | 13.7 | 12.8 | 12.9 | 12.9 |
| **JJA** | Mean (µg m$^{-3}$) | 63.6 | 78.6 | 80.8 | 84.4 | 85.6 |
| | **Difference in model mean (%)** | | -2.8 | | -1.4 | |
| | NMB (%) | | 23.7 | 27.1 | | |
| | SD (µg m$^{-3}$) | 10.2 | 16.3 | 15.1 | 20.6 | 20.5 |
| **SON** | Mean (µg m$^{-3}$) | 46.3 | 48.6 | 55.0 | 52.7 | 54.9 |
| | **Difference in model mean (%)** | | -13.2 | | -4.2 | |
| | NMB (%) | | 4.9 | 18.8 | | |
| | SD (µg m$^{-3}$) | 10.2 | 15.0 | 14.2 | 15.2 | 14.1 |

**Table 2: Statistical results comparing seasonal mean PM$_{2.5}$ concentrations simulated at the coarse and finer resolutions to observations from 25 stations within the EMEP network in 2007. Statistical results for all model grid-cells of both resolutions are also shown. Percentage differences between the two model resolutions are calculated as (PM$_{2.5 \text{ coarse resolution}}$ − PM$_{2.5 \text{ finer resolution}}$)/(PM$_{2.5 \text{ coarse resolution}}$).**

| Season | | Obs. | 25 sites Model | | All grid-cells Model | |
|---|---|---|---|---|---|---|
| | | | 140 km | 50 km | 140 km | 50 km |
| DJF | Mean (µg m$^{-3}$) | 12.1 | 8.3 | 9.5 | 5.1 | 5.5 |
| | **Difference in model mean (%)** | | -14.5 | | -7.8 | |
| | NMB (%) | | -31.0 | -21.3 | | |
| | SD (µg m$^{-3}$) | 9.2 | 2.5 | 3.1 | 3.1 | 3.7 |
| MAM | Mean (µg m$^{-3}$) | 12.6 | 12.4 | 16.2 | 9.0 | 9.5 |
| | **Difference in model mean (%)** | | -30.6 | | -5.5 | |
| | NMB (%) | | -0.2 | 31.1 | | |
| | SD (µg m$^{-3}$) | 5.1 | 2.6 | 5.4 | 4.9 | 6.2 |
| JJA | Mean (µg m$^{-3}$) | 10.6 | 18.0 | 14.9 | 11.9 | 8.4 |
| | **Difference in model mean (%)** | | 17.2 | | 29.4 | |
| | NMB (%) | | 70.0 | 40.1 | | |
| | SD (µg m$^{-3}$) | 4.0 | 5.4 | 6.4 | 7.0 | 6.2 |
| SON | Mean (µg m$^{-3}$) | 11.0 | 10.7 | 13.2 | 12.3 | 11.3 |
| | **Difference in model mean (%)** | | -23.4 | | 8.1 | |
| | NMB (%) | | -2.4 | 22.0 | | |
| | SD (µg m$^{-3}$) | 4.8 | 4.1 | 10.3 | 7.0 | 6.7 |

**Table 3: Warm season (April-September) mean of daily maximum 8-hour running mean O$_3$ concentrations (MDA8 O$_3$) and annual mean PM$_{2.5}$ concentrations at the coarse and finer resolutions compared to observations from 52 and 25 stations within the EMEP network, respectively.**

| Season | | Obs. | 140 km | 50 km |
|---|---|---|---|---|
| **MDA8 O$_3$ (Apr - Sept)** | Mean (µg m$^{-3}$) | 86.3 | 95.6 | 94.8 |
| | **Difference in model mean (%)** | | 0.8 | |
| | NMB (%) | | 10.9 | 8.9 |
| | SD (µg m$^{-3}$) | 9.2 | 14.7 | 14.2 |
| **PM$_{2.5}$ (Annual)** | Mean (µg m$^{-3}$) | 11.4 | 12.6 | 13.7 |
| | **Difference in model mean (%)** | | -8.7 | |
| | NMB (%) | | 10.5 | 20.2 |
| | SD (µg m$^{-3}$) | 5.1 | 2.8 | 5.0 |





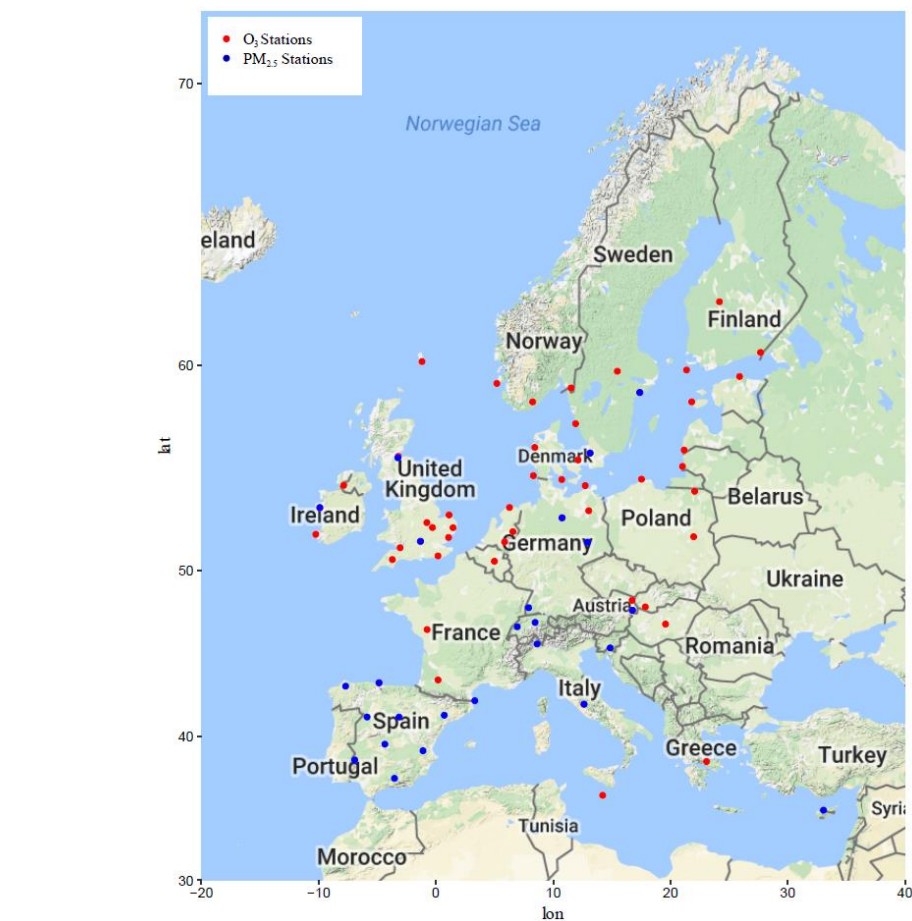

**Figure 1: EMEP measurement stations with altitude less than or equal to 200 m, used for seasonal mean surface O₃ comparison to modelled concentrations (52 sites – red) and EMEP measurement stations used for seasonal mean PM₂.₅ comparison to modelled concentrations (25 sites - blue)**

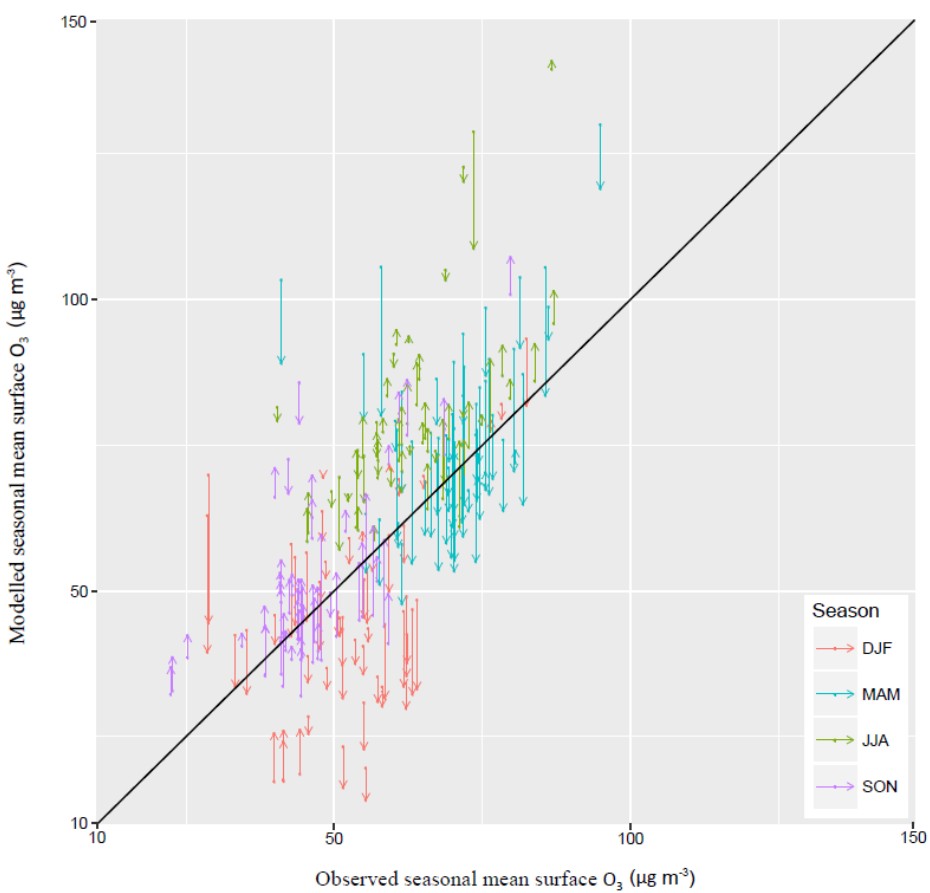

**Figure 2: Seasonal mean modelled vs observed O₃ for 52 sites across the EMEP network for the year 2007. The arrow tails mark O₃ concentrations at the coarse resolution while the arrow heads represent the corresponding O₃ concentrations at the finer resolution. The 1:1 line shows agreement between observed and simulated O₃.**

835

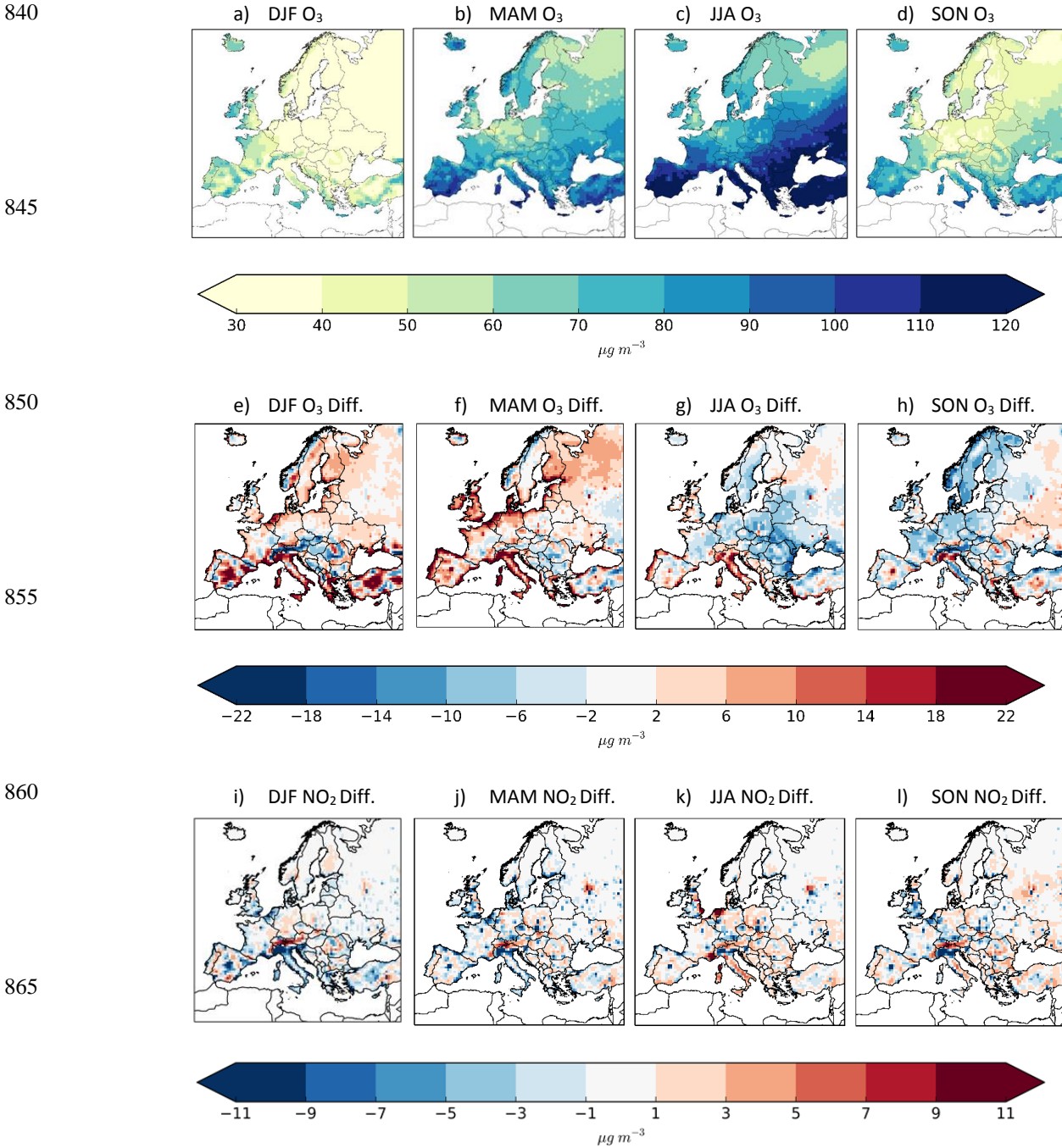

**Figure 3: Seasonal mean O₃ simulated at the finer resolution (top panel), differences in seasonal mean O₃ between the coarse and finer resolutions (O₃ coarse resolution − O₃ finer resolution) (middle panel) and NO₂ (NO₂ coarse resolution − NO₂ finer resolution) (bottom panel). Blue regions in middle and bottom panels indicate that pollutant concentrations at the coarse resolution are lower (negative difference) while red regions indicate that concentrations are higher (positive difference) than those at the finer resolution.**

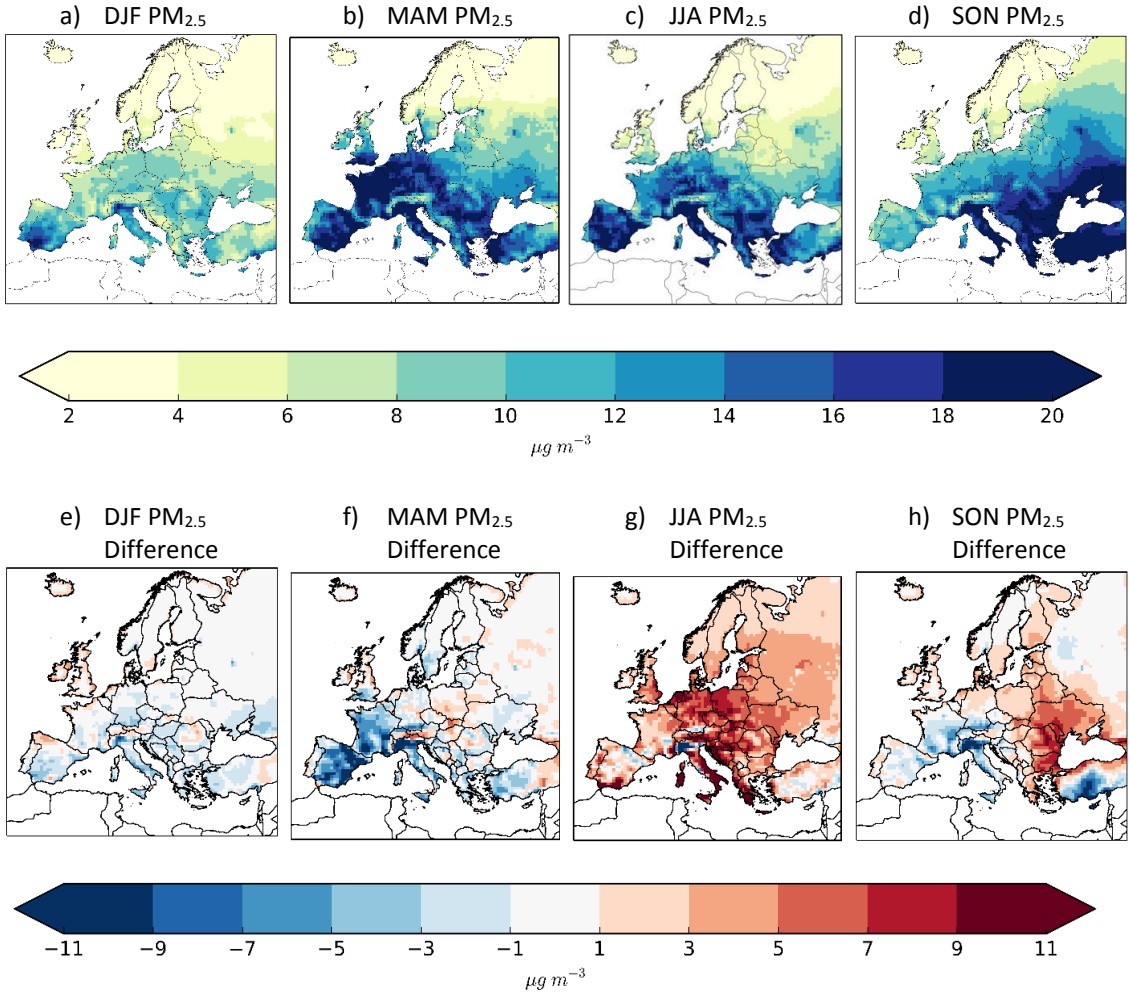

875

**Figure 4: Seasonal mean PM₂.₅ simulate at the finer resolution (top panel) and differences between seasonal mean PM₂.₅ at the coarse and finer resolution in 2007 (PM₂.₅ coarse resolution − PM₂.₅ finer resolution) (bottom panel).**

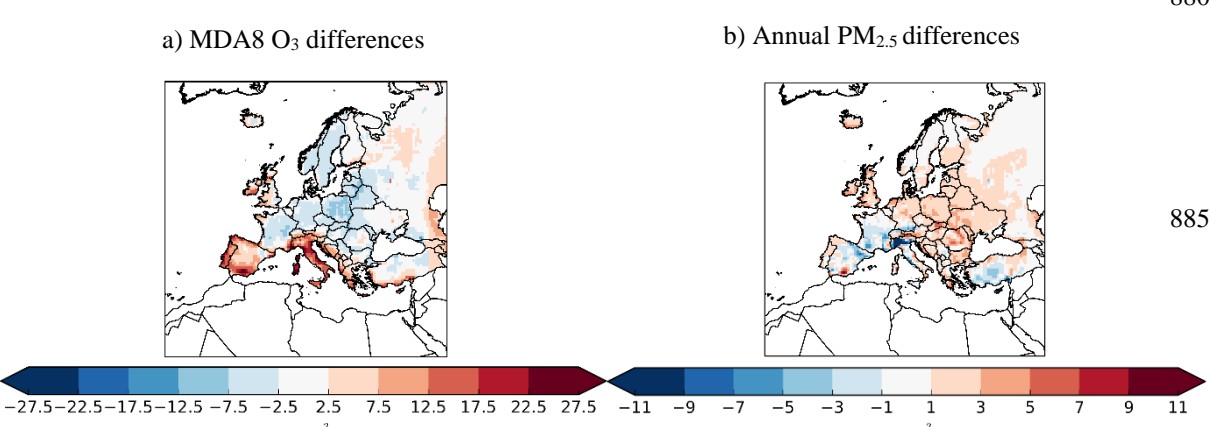

**Figure 5: Differences in a) warm season (April-September) mean of daily maximum 8-hour running mean O₃ (concentrations above 70 μg m⁻³) and b) annual mean PM₂.₅ between the coarse and finer resolution (coarse – finer).**

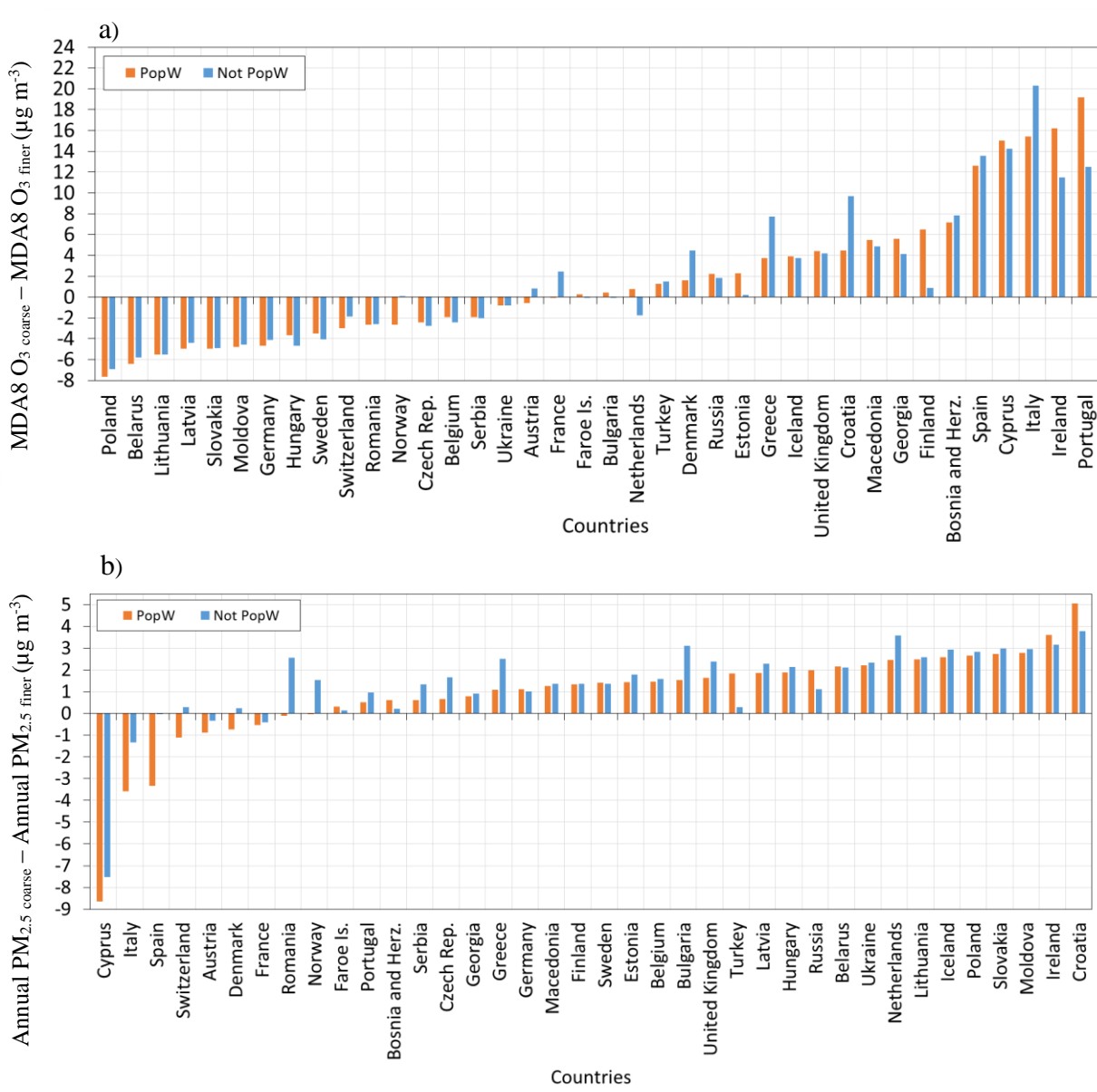

**Figure 6: a) Differences between warm season mean daily maximum 8-hour running mean (MDA8) O₃ concentrations simulated at the two resolutions (coarse – finer) for population-weighted (PopW) concentrations (orange bars) and concentrations with no population-weighting (blue bars) b) same holds for annual mean PM₂.₅ concentrations. Countries are ordered by differences in PopW pollutant concentrations between the two resolutions.**

890

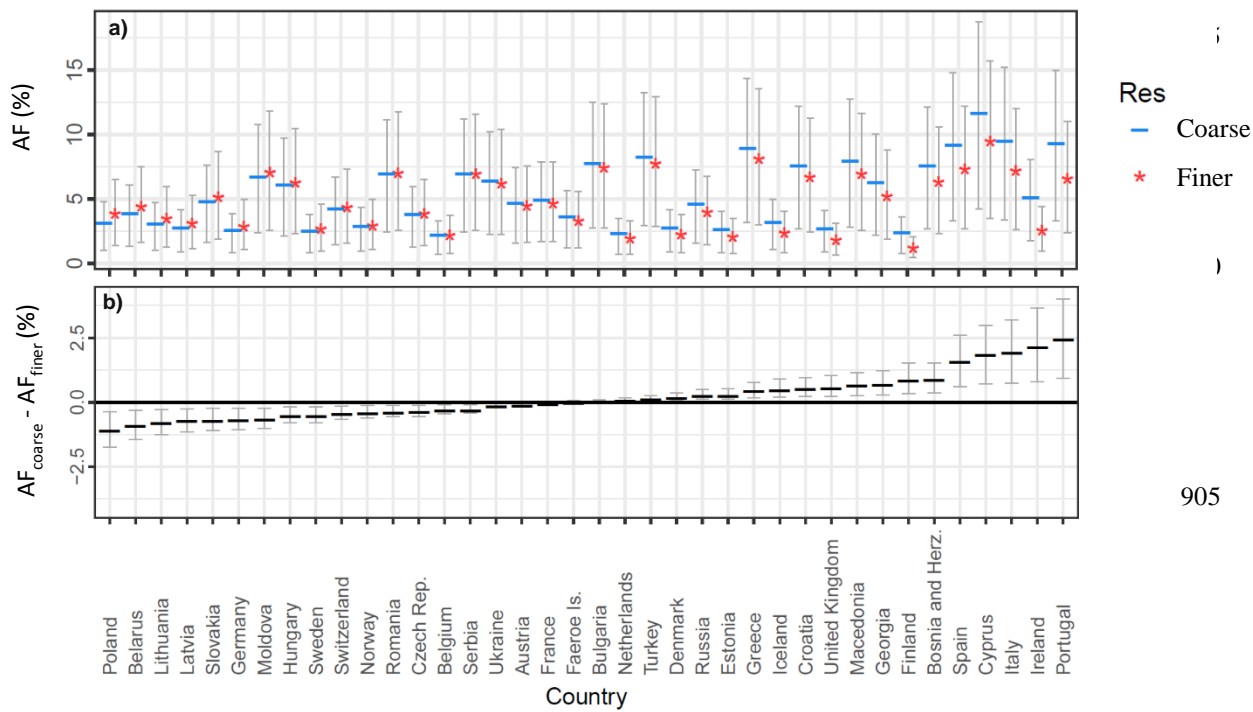

**Figure 7: a) AF associated with long term exposure to daily maximum 8-hour running mean O₃ for each model resolution expressed as a percentage b) Differences in AF between the two resolutions expressed as a percentage for each European country (AF_coarse − AF_finer). Grey lines show the 95% C.I. which represents uncertainties associated only with the concentration-response coefficient used.**

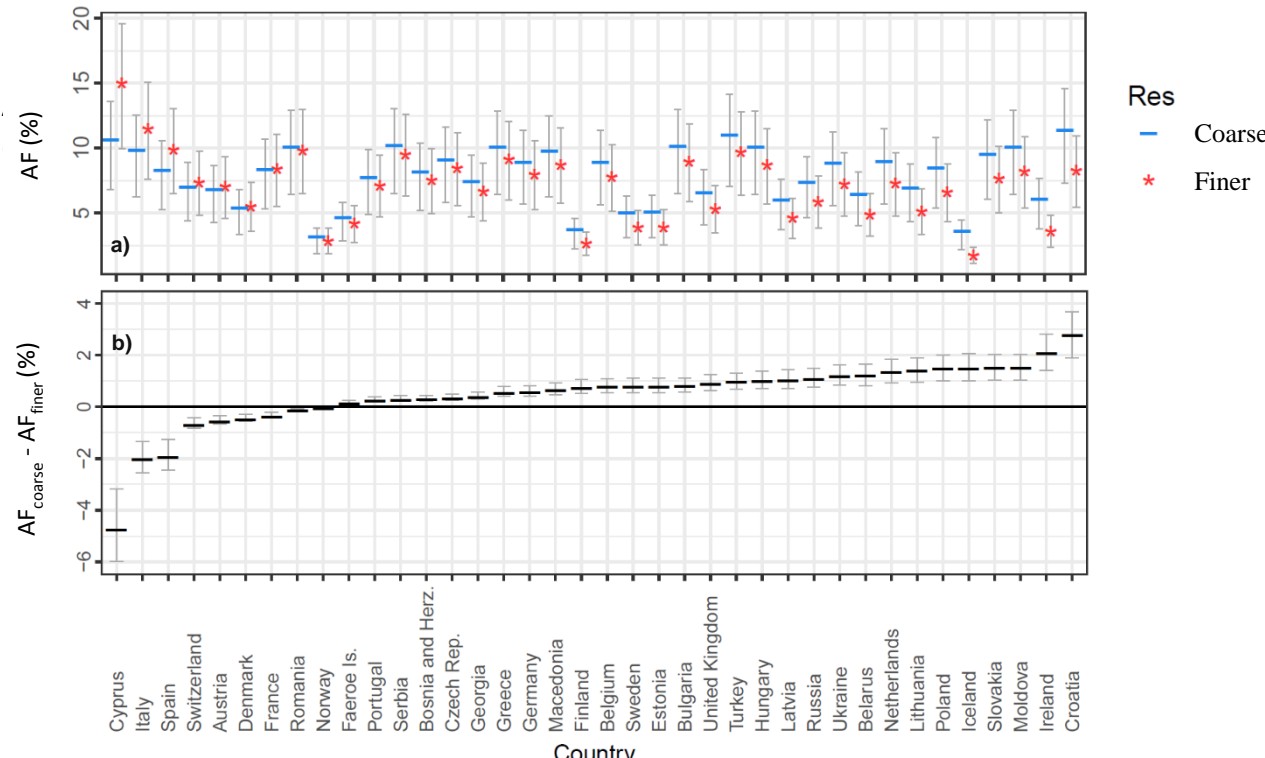

**Figure 8: a) AF associated with long-term exposure to PM₂.₅ for each model resolution expressed as a percentage b) Differences in AF between the two resolutions expressed as a percentage for each European country (AF_coarse − AF_finer). Grey lines show the 95 % C.I. which represents uncertainties associated only with the concentration-response coefficient used.**