# Peer review of "The influence of model spatial resolution on simulated ozone and fine particulate matter for Europe: implications for health impact assessments"

_Atmospheric Chemistry and Physics, 2017_

## Referee Comment (RC1) · Anonymous Referee #1 · 13 Dec 2017

This study compares health impacts estimated for ozone and PM2.5 simulated at global versus regional chemical transport model resolutions, and analyzes the factors contributing to resulting differences in the health estimates. While several other studies have conducted similar analyses for air pollution health impacts in the US, at a range of spatial scales from 4km to ∼250 km, it hasn't been done for Europe. There is reason to believe that results from the US wouldn't directly apply to Europe due to differences in emissions magnitudes of pollution components and chemical processing in the atmosphere. Thus, while this paper is a relatively straightforward corollary to the US studies, it is interesting and a useful contribution to the literature. It also presents some interesting new results on seasonality of and factors contributing to the resolution ef-

fect.

Below are some comments for the authors to consider: - In reality, differences in disease rates in urban centers versus broader areas would also come into play in addition to model spatial resolution and population/pollution colocation. Please discuss this limitation in your approach and how its omission should affect interpretation of your results.

- The paper made me wonder why so many papers have been written on spatial resolution issues, but not as much attention has been given to vertical resolution. Is the use of the first model layer (which is noted as 40m high) as ground concentrations adequate for capturing concentrations at the "nose level"? Should this be explored? Please provide guidance on this issue.

- Line 160-167: The original PM2.5 epi study should be cited here, rather than simply referring to the HRAPIE recommendations. Please check whether all the health impact assessments referenced (Anenberg 2009, Punger and West 2013, and Thompson et al. 2014) actually used HRAPIE recommended effect estimates since some of these were published before HRAPIE.

- Many health impact assessments are now employing non-linear concentration-response curves which flatten out considerably at higher concentrations, particularly for cardiovascular diseases. Please comment on how using such non-linear concentration-response functions would influence your results (e.g if the higher spatial resolution leads to higher PM concentrations, would those concentrations then fall on the flatter end of the CRF, leading to lower health impact estimates?)

- Same comment as above, but for low-concentration thresholds. We don't know whether PM health effects go down to zero, though some epi studies are showing relationships to very clean levels (2-5 ug/m3). It's useful to the reader to provide some guidance on how your results would be different if you did apply a low-concentration threshold for PM2.5, perhaps set at the theoretical minimum risk level used in the GBD

studies.

- Also, the most recent American Cancer Society study update gives ozone-mortality relationships for annual average concentrations (Turner et al. Long-term ozone exposure and mortality in a large prospective study, American Journal of Respiratory and Critical Care Medicine, 193, 10, 1142, 2015). These relationships were used by Malley et al. to updated the ozone burden of disease values (Malley et al. Updated global estimates of respiratory mortality in adults >30 years of age attributable to long-term ozone exposure, Environmental Health Perspectives, 087021-1, 2017). Please comment on how your results would be different if you were to use these annual average ozone effect estimates, given the seasonality of the resolution effect on simulated concentrations.

- Line 183: GPW data are at a much finer resolution. Were these regridded to 0.5x0.5 degrees?

- Lines 426-433: should compare results also to Thompson et al. 2014, and also compare results for ozone from these studies.

---

## Referee Comment (RC2) · Anonymous Referee #2 · 4 Feb 2018

The manuscript by Fenech et al. considers the impact of model resolution (140 km vs 50 km) on the attributable fraction of premature mortality to O3 and PM2.5 in Europe. This question of model resolution influences on such health effects estimates has been raised previously in a few other targeted studies but has yet to be evaluated in Europe at these scales. The authors find that the impact of resolution is spatially variable, and significant. Hence, this study is of value of the community for better understanding health impact assessments in Europe, and contributes more broadly to a body of work that helps us understand the mechanisms governing scale dependencies. The manuscript is clearly organized and easy to read. There are through some areas where the analysis could be more focused, and I have some concerns related

to model performance at the two different resolutions, and how that translates into a potential recommendation for future research into health impacts. These aspects and others are described in detail below; addressing them will constitute minor revisions, after which this paper will be suitable for publication in ACP.

Major comments:

156: I understand the authors motivation here, to isolate the impact of model resolution from the impact of resolving differences in baseline mortalities. However, I disagree with their approach. But computing country-level AF and country-level baseline mortalities, the authors neglect any impact on mortality estimates that may come from sub-national variability in AF and baseline mortalities. It seems to me that a better (?) approach would be to map the O3 and PM2.5 concentrations from both the coarse and fine simulations to the same fine-scale resolution of the available population and baseline mortality rate information. This way they would have a consistent comparison that isolates the impact of the air quality model resolution, but their final estimates of mortality would be more accurate and more sensitive to differences in the air quality model resolutions. I'd suggest they at least consider this approach, which is just a post-processing step and doesn't involve any more model simulations, to see if it makes a significant difference, or explain why it isn't the recommended approach.

When presenting the AF results, it would be interesting to know if the differences between the two scales of analysis are greater than the error bars in the AF estimates stemming from the uncertainty in the concentration response parameter (beta). In other words, when are the model-dependent differences significant, compared to the health-data uncertainties? See papers by Thompson et al. in this regard.

It seems somewhat problematic, in terms of drawing conclusions, that the fine-scale simulated concentrations are, in many seasons, a poorer match to the observations than the coarse scale simulations, for both O3 and PM2.5. I strongly insist that the authors should present the statistical evaluation of biases in O3 during the warm season and annual ave PM2.5 in the main text, not the supplemental, as these are the scales most relevant to the focus of this work (health impacts). This is rather critical information that the reader shouldn't have to dig for.

Overall, given these biases, would the authors recommend using the fine scale model over the coarse scale model for health impact analysis, especially for PM2.5 where the bias in the annual average concentration is higher at the finer scale? Or are there enough observations to say which is better at estimating exposure? This wasn't clear to me. I think this warrants some discussion, with conclusions in the abstract and conclusions.

Also, model bias relative to observations should be considered when discussing regional differences in modeled spatial resolution of population-weighted concentrations (section 4.2) – in other words, is one model resolution notably better in heavily populated areas? This is a critical question which I couldn't find a direct evaluation of, although all the pieces are available to make the comparison. The same comment applies to comparison of AF (section 4.3).

426 - 437: Regarding comparison to studies in the US, I think an interesting conclusion is that the differences owing to model resolution is not something that is consistent in sign, spatially (or that could thus be easily corrected for without knowing the spatial dependence). This is self consistent with their own evaluation of the variability of the difference across regions within Europe. Still, one might hypothesize about additional factors that control these differences. Did the authors consider the speciation of the PM2.5 and how this might affect the differences between coarse and fine scale simulations? For example, both Punger and West (2013) and Li et al. (2015) note that the differences are more significant for primary anthropogenic PM (e.g., BC) than secondary anthropogenic PM or primary natural PM. I contrast, Thompson 2014 noted the biggest impact of resolution going from 36 km to 4 km was for secondary PM. I didn't see PM2.5 speciation discussed anywhere in the present work.

Minor comments:

1.13: Given that it is a regional modeling study, would make more sense to phrase as "at resolutions typical of global (∼140 km) and regional (∼50km) models." Throughout, it would make more sense to me if the results were referred to as "coarse" vs "fine" rather than "global" vs "regional", all results are regional and this could be misleading to someone just glancing at the figures. Further, most regional models these days are more like 12 km scale or finer.

1.15: The differences seem a bit more modest, all less than 30% and most less than 10%. Not sure if "strong" is the right word.

28-33: Readers not familiar with AF may think these numbers are very small – the authors might wish instead (or additionally) to present the amount by which AF is changing owing to model resolution (i.e. a factor of two to three). So consider not the changes in total mortality (which is small, 5%) but instead the % changes in pollution attributed mortality. I think the authors should also state the differences in the total over the entire domain, rather than just the range across regions, even if the total benefits from some fortuitous cancelation of under and over estimates.

52-62: I'm not sure how "strong" the effects are that are being discussed in this paragraph – can the authors be more quantitative when describing previous works? The following paragraph on PM2.5 is much better in this regards.

68: The reference is Li et al. (2015). not 2016. The authors refer to the paper both ways in the text but only include an incorrect reference to 2016 in the bibliography.

78: The authors should also consider the results of Thomson and Selin (2012), which found there were some differences between O3-related premature deaths at the 36 km scale and finer (24, 12, and 4 km) scales, although these tended to lie within the range of uncertainty of the health impact estimates. Further, they should discuss the O3 health impact results from Thompson 2014, which were found to be more sensitive

to resolution than PM2.5 health impacts.

Table 1: The placement of the "difference" row is confusing. It is the difference in the model simulated mean, and should be labeled as such and more clearly located directly below the row reporting the mean, not the rows reporting NMB and SD. The choice of significant figures for the difference also seems odd. For example, why the DJF mean difference is reported as 10 rather than 9.6 isn't clear, when other numbers are resolved to the tenth of a ug/m3. Same comments apply to Table 2. Annual average PM2.5 and warm-season (April - Sept) O3 should be added to these tables, not put in the SI.

[Figure]

---

## Author Comment (AC1) · 22 Mar 2018

Dear Dr Duncan,

We thank both reviewers for their insightful comments that have aided the improvement of this manuscript considerably. Our responses to the reviewers' comments together with a description of the changes made to the manuscript can be found below. Changes not mentioned in this document are purely editorial. For clarity, the reviewers' comments are copied below in bold, followed by our responses, and in quotes and italics, modifications to the manuscript. In our revised manuscript, the modified text is shown using track changes.

**Referee #1**

**This study compares health impacts estimated for ozone and PM$_{2.5}$ simulated at global versus regional chemical transport model resolutions, and analyzes the factors contributing to resulting differences in the health estimates. While several other studies have conducted similar analyses for air pollution health impacts in the US, at a range of spatial scales from 4km to _250 km, it hasn't been done for Europe. There is reason to believe that results from the US wouldn't directly apply to Europe due to differences in emissions magnitudes of pollution components and chemical processing in the atmosphere. Thus, while this paper is a relatively straightforward corollary to the US studies, it is interesting and a useful contribution to the literature. It also presents some interesting new results on seasonality of and factors contributing to the resolution effect.**

We thank the reviewer for their positive comments.

**Comments:**

**In reality, differences in disease rates in urban centers versus broader areas would also come into play in addition to model spatial resolution and population/pollution colocation. Please discuss this limitation in your approach and how its omission should affect interpretation of your results.**

We agree with the reviewer that disease rates in urban centres versus broader areas could vary and thus may need to be taken into consideration when studying regional to urban-scale effects of air pollution on morality. However we do not have this information. More importantly, in our study we did not examine changes in absolute mortality attributable to long-term exposure

to $O_3$ and $PM_{2.5}$ but we examined changes in the Attributable Fraction (AF) of all-cause mortality to isolate the impact of changing the resolution on pollutant concentrations and the associated health impacts, from changes in baseline mortality rates across different countries. AF represents the fraction or percentage of the all-cause mortality which is attributable to the effects of $O_3$ and $PM_{2.5}$, and depends only on population weighted pollutant concentration and an appropriate concentration-response coefficient which is typically applied at a country or continental-scale level (e.g. see WHO 2013).

**The paper made me wonder why so many papers have been written on spatial resolution issues, but not as much attention has been given to vertical resolution. Is the use of the first model layer (which is noted as 40m high) as ground concentrations adequate for capturing concentrations at the "nose level"? Should this be explored? Please provide guidance on this issue.**

We thank the reviewer for this comment and acknowledge that using concentrations at lowest model level (with a height of 40 m) is a limitation of our study. The lowest model level is widely used as representative of surface concentrations in modelling studies and simulated concentrations are evaluated against measurements, but some studies e.g. Fiore et al. (2009) note uncertainties pertaining to vertical resolution in coarse global-scale models. Similarly, the lowest model layer is used when calculating health impact assessments (e.g. Punger and West, 2013). For pollutants with an extremely short lifetime such as $NO_2$ vertical resolution could be a very important issue but less so for longer lived $O_3$ and $PM_{2.5}$ investigated in this study. We have evaluated the model output using observations at ground level and found performance to be satisfactory (Sections 3.1 and 3.3) We have added the following text to the methods section 2.1 and have added a note to the conclusions in Section 5:

Page 5, line 148: "*All pollutant concentrations used in this study have been extracted at the lowest model level with a mid-point at 20 m. While this level is considered representative of surface or ground- level concentrations, local orographically driven flows or sharp gradients in mixing depths cannot be represented at this vertical resolution (Fiore et al. 2009).*"

Page 19, line 593: "*The pollutant concentrations used in this study have been extracted at the lowest model level with a mid-point at 20 m. The sensitivity of our simulated pollutant concentrations to vertical model resolution has not been examined.*"

**Line 160-167: The original PM2.5 epi study should be cited here, rather than simply referring to the HRAPIE recommendations. Please check whether all the health impact assessments referenced (Anenberg 2009, Punger and West 2013, and Thompson et al. 2014) actually used HRAPIE recommended effect estimates since some of these were published before HRAPIE.**

We thank the reviewer for this comment. We have modified the text to clarify the sources of the concentrations response coefficients related to long-term exposure to $O_3$ and $PM_{2.5}$ used in our study and in the other studies cited (Section 2.3):

Page 6 line 173: "Although there is limited evidence available for the long-term health impacts of $O_3$ especially in Europe (The UK Committee on the Medical Effects of Air Pollution (COMEAP) 2015), *a number of studies have quantified the adverse health impacts associated with long-term exposure to $O_3$. In this study we apply the Health Risks of Air Pollution in Europe – HRAPIE project recommended coefficient for long-term exposure to $O_3$* (WHO, 2013) *to investigate the sensitivity of health calculations to the model resolution used to simulate $O_3$ concentrations. This concentration–response coefficient is derived from the single-pollutant analysis of the American Cancer Society Cancer Prevention Study II (CPS II) cohort study data in 96 metropolitan areas of the US* (Jerrett et al., 2009) *which has been used by previous studies (e.g.* Anenberg et al., 2009; Punger and West, 2013; Thompson et al., 2014; Cohen et al., 2017); *but is re-scaled from 1-hour mean to 8-hour mean concentrations using the ratio 0.72, derived from the APHEA-2 project* (Gryparis et al., 2004). *The value recommended by HRAPIE for the concentration-response coefficient, or β value (Eq.1), for the effects of long-term $O_3$ exposure on respiratory mortality recommended is 1.014 (95% Confidence Interval (CI) = 1.005, 1.024) per 10 µg m$^{-3}$ increase in MDA8 $O_3$ during the warm season (April-September) with a threshold of 70 µg m$^{-3}$* (WHO, 2013). *For estimating the health impact of long-term exposure to $PM_{2.5}$ on all-cause (excluding external) mortality, HRAPIE (WHO 2013) recommends a relative risk coefficient of 1.062 (95% CI = 1.040, 1.083) per 10 µg m$^{-3}$ increase in annual average concentrations (with no threshold) which is based on a meta-analysis of cohort studies by* Hoek et al. (2013)."

**Many health impact assessments are now employing non-linear concentration-response curves which flatten out considerably at higher concentrations, particularly for cardiovascular diseases. Please comment on how using such non-linear concentration-**

**response functions would influence your results (e.g if the higher spatial resolution leads to higher PM concentrations, would those concentrations then fall on the flatter end of the CRF, leading to lower health impact estimates?)**

To estimate the global burden of disease attributable to ambient fine particulate matter exposure, some recent studies have derived integrated concentration-response functions that come from integrating available relative risk information from various studies of ambient air pollution, second hand tobacco smoke, household solid cooking fuel and active smoking (e.g. Burnett et al., 2014). These functions are applied to cause-specific mortality associated with long-term exposure to $PM_{2.5}$, however in this study we have focused on all-cause $PM_{2.5}$-related mortality. We agree that by using the 'integrated' concentration-response function, the concentration response curves flatten out at high concentrations based on evidence from epidemiological studies. However for ambient air pollution the curve is log-linear (e.g. Fig. 1 and 2 Burnett et al., 2014). In addition the curve flattens out for annual $PM_{2.5}$ concentrations above approximately 100 µg m$^{-3}$ (Burnett et al., 2014). Such high annual ambient $PM_{2.5}$ concentrations are common in cities across Asia and other developing regions (Brauer et al., 2012; Health Effects Institute, 2010). However, in our study across the whole European domain, the maximum annual mean $PM_{2.5}$ concentrations are 40 µg m$^{-3}$ and 49 µg m$^{-3}$ for the global and regional configuration, respectively. Given the magnitude of the concentrations in this Europe focused study we feel that applying a log-linear relationship is appropriate.

**Same comment as above, but for low-concentration thresholds. We don't know whether PM health effects go down to zero, though some epi studies are showing relationships to very clean levels (2-5 ug/m3). It's useful to the reader to provide some guidance on how your results would be different if you did apply a low-concentration threshold for PM2.5, perhaps set at the theoretical minimum risk level used in the GBD studies.**

We thank the reviewer for this comment, and we have now investigated the impact of a low concentration threshold. We apply a threshold of 5.8 µg m$^{-3}$ (following the minimum that is suggested by Burnett et al. (2014) which is derived from Lim et al. (2012)). We find differences in AF estimates associated with long-term exposure to population-weighted $PM_{2.5}$ range from -4.8% to +2.1% compared to -4.7% to +2.8% when no threshold is applied. The spatial distribution of these estimates remains unchanged with a large number of countries in Eastern Europe and the UK showing positive differences in AF between the global and regional

resolutions and only slight changes in country rankings (see Fig. R1 below compared to Fig.8b in manuscript). Hence, in our study we find the effect of applying a low-concentration threshold for PM$_{2.5}$ to be small. We have added the following text to the manuscript to discuss these results in section 4.4 and added a sentence to the conclusions in Section 5.

Page 16, line 502: *"We also examine the impact of using a low-concentration threshold. We apply a threshold of 5.8 µg m-3 (suggested by Burnett et al. (2014) which is derived from Lim et al. (2012)) to annual mean PM$_{2.5}$ concentrations. Differences in AF estimates associated with long-term exposure to population-weighted PM$_{2.5}$ concentrations range from -4.8% to +2.1% (as compared to -4.7% to +2.8% above when no threshold is applied). The spatial distribution of these estimates remains unchanged and only slight changes in country rankings occur. Hence, the impact of applying a low concentrations threshold in this study for Europe is small."*

Page 19 line 583: *"In addition, these ranges in AF associated with long-term exposure to annual mean PM2.5 were largely unaltered with the application of a low-concentration threshold for PM$_{2.5}$."*

[Figure]

**Figure R1:** Differences in AF associated with long-term exposure to annual mean PM$_{2.5}$ between the two resolutions expressed as a percentage for each European country (AF$_{global}$ − AF$_{regional}$) using a threshold of 5.8 µg m$^{-3}$. Grey lines show the 95 % C.I. which represents uncertainties associated only with the concentration-response coefficient used.

**Also, the most recent American Cancer Society study update gives ozone-mortality relationships for annual average concentrations (Turner et al. Long-term ozone exposure and mortality in a large prospective study, American Journal of Respiratory and Critical Care Medicine, 193, 10, 1142, 2015). These relationships were used by Malley et al. to updated the ozone burden of disease values (Malley et al. Updated global estimates of respiratory mortality in adults >30 years of age attributable to long-term ozone exposure, Environmental Health Perspectives, 087021-1, 2017). Please comment on how your results would be different if you were to use these annual average ozone effect estimates, given the seasonality of the resolution effect on simulated concentrations.**

We thank the reviewer for this interesting comment and we have investigated this effect. When using the concentration-response function (CRF) based on epidemiological studies, the time averaging period used for pollutant concentrations should match that used to quantify the CRF. For this reason we do not use annual mean MDA8 $O_3$ concentrations in conjunction with the CRF used in our study (based on HRAPIE in turn based on Jerrett et al. 2009) as this was derived from warm season concentrations. Thus, in addition, we have estimated the differences in AF between the two resolutions following Turner et al. (2015) whereby we use: a) an annual-mean MDA8 $O_3$ concentration (instead of summer mean concentrations), b) a concentration response function of 1.06 (95% CI: 1.04-1.08) per 10 µg m$^{-3}$ (instead of 1.014 (95% CI: 1.005-1.024) per 10 µg m$^{-3}$ and c) a threshold of 53.4 µg m$^{-3}$ (instead of 70 µg m$^{-3}$), with the values in parenthesis being those used to date in our study. Results are shown in Fig. R2 below (compare to Fig. 7 in manuscript).

Differences in AF between the two resolutions using annual-mean $O_3$ concentrations and CRF/threshold values from Turner et al. (2015) range from -2.3% to +12.0% across the countries compared to -0.9% to +2.6% when a summer mean MDA8 $O_3$ concentration with the CRF from the WHO HRAPIE project is used. The CRF quoted by Turner et al. (2015), applicable to annual-mean $O_3$, is approximately 4 times higher than the CRF used in our study which is derived from summer time MDA8 $O_3$ exposure. This is the main driver for a larger range in differences in AF between the resolutions when using the recommendations in Turner et al. (2015) for annual MDA8 $O_3$ concentrations.

In contrast, Turner et al. (2015) found similar results for $O_3$-mortality relationships for all-cause mortality, diseases of the circulatory system and cerebrovascular diseases when using

summer and annual-mean $O_3$ concentrations. However results were attenuated when using a summer $O_3$ concentration for mortality due to dysrhythmias, heart failure, and cardiac arrest, diabetes and respiratory causes. Although as discussed previously we do not suggest applying the HRAPIE coefficient to annual-average concentrations, we have done this calculation as a sensitivity test (Fig. R3). When using the HRAPIE suggested coefficient derived from Jerrett et al. (2009) with annual MDA8 $O_3$ concentrations, differences in AF range from -0.5% to +3.7%. The range is slightly larger compared to the summer mean estimates as differences in annual mean MDA8 $O_3$ concentrations between the two resolutions are larger due to the seasonality noted in the Section 3 of the manuscript and as mentioned in the conclusion section.

We have added these results to our manuscript through the following text in section 4.3 and section 5 conclusions:

Page 15, line 474: *"Since, seasonal differences in simulated $O_3$ with resolution are considerable, the AF associated with long-term exposure to $O_3$ was also calculated based on annual-mean (as opposed to summer-mean) $O_3$ concentrations based on recommendations by Turner et al. (2015). Turner et al (2015) suggest a higher concentration response coefficient of 1.06 (95% CI: 1.04-1.08) per 10 µg m$^{-3}$ and a slight lower MDA8 $O_3$ threshold of 53.4 µg m$^{-3}$ compared to values used in our study for summer-mean MDA8 $O_3$. Using the values from Turner et al. (2015) the differences in AF are found to be of the same sign for the majority of the countries and the rankings across countries are largely similar. This similarity occurs because the difference in annual-mean MDA8 $O_3$ concentrations between the two resolutions shows generally similar spatial patterns to the differences in warm season MDA8 $O_3$ concentrations (not shown). However the ranges when using annual-mean $O_3$ concentrations and recommendations form Turner et al. (1015) are larger: -2.3% to +12.0%, compared to AF ranges given above for MDA8 $O_3$. From further sensitivity analyses it is found that these greater AF ranges can be attributed to the use of a higher concentration-response coefficient (by a factor of approximately 4) rather than differences in annual-mean compared to summer-mean concentrations.*

Page 19 line 586: *"When using annual-mean MDA8 $O_3$ concentrations alongside a recommended concentration-response coefficient and threshold suggested by Turner et al. (2015) the difference in AF between the two resolutions is considerably larger than our estimates using summer-mean MDA8 $O_3$ concentrations. This is driven by the higher concentration-response coefficient (by a factor of approximately 4) quoted in Turner et al.*

*(2015) compared to that suggested by HRAPIE for summer mean MDA8 O₃ concentrations (WHO, 2013)."*

[Figure]

**Figure R2:** Differences in AF associated with long-term exposure to annual mean MDA8 O₃ between the two resolutions expressed as a percentage for each European country (AF_global − AF_regional) using a threshold of 53.4 µg m⁻³. Grey lines show the 95 % C.I. which represents uncertainties associated only with the concentration-response coefficient quoted in Turner et al. (2015).

[Figure]

**Figure R3:** Differences in AF associated with long-term exposure to annual mean MDA8 $O_3$ between the two resolutions expressed as a percentage for each European country ($AF_{global} - AF_{regional}$) using a threshold of 70 µg m$^{-3}$. Grey lines show the 95 % C.I. which represents uncertainties associated only with the concentration-response coefficient quoted in HRAPIE which is based on Jerrett et al. (2009).

**- Line 183: GPW data are at a much finer resolution. Were these regridded to 0.5x0.5 degrees?**

We thank the review for pointing this out. The GPW were summed up to produce the total population that falls within each chemistry-climate model grid cell. To clarify this point we have edited the manuscript as follows.

Page 7 Line 215: "*Here, $x_i$ represents the pollutant concentration within each model grid-cell i and $p_i$ represents the total population (aged 30+ years) summed within each model grid-cell.*"

**- Lines 426-433: should compare results also to Thompson et al. 2014, and also compare results for ozone from these studies.**

We thank the reviewer for this point, although we do note the difficulty of definitive comparisons as all the studies we compare our results to are for the USA. To make clear that our study region is Europe we have added 'for Europe' to the title of the manuscript. We have also added the following text comparing our $O_3$ and $PM_{2.5}$ results to these U.S. based studies, respectively.

Page 15 line 465: *"For U.S. averaged mortality estimates, Punger and West (2013) show that mortality estimates related to long-term $O_3$ exposure, calculated using the $O_3$ concentrations at 36 km, were higher (by 12%) than estimates calculated at the 12 km resolution. Resolution was also found to play and important role in determining health benefits associated with differences in $O_3$ between 2005 and 2014 in the U.S. (Thompson et al. 2014). In particular, in urban areas, Thompson et al. (2014) estimate that the benefits calculated using coarse resolution results were on average two times greater than estimates calculated using the finer scale results. Both the studies mentioned are conducted in the U.S. and use a different concentration response coefficient and thus a definitive comparison between these studies and our estimates over Europe is not possible."*

Page 17 line 512: *"In contrast, Thompson et al. (2014) find that health benefits associated with changes in $PM_{2.5}$ concentrations between 2005 and 2014 in the U.S., were not sensitive to resolution."*

**Referee #2**

The manuscript by Fenech et al. considers the impact of model resolution (140 km vs 50 km) on the attributable fraction of premature mortality to O3 and PM2.5 in Europe. This question of model resolution influences on such health effects estimates has been raised previously in a few other targeted studies but has yet to be evaluated in Europe at these scales. The authors find that the impact of resolution is spatially variable, and significant. Hence, this study is of value of the community for better understanding health impact assessments in Europe, and contributes more broadly to a body of work that helps us understand the mechanisms governing scale dependencies. The manuscript is clearly organized and easy to read. There are through some areas where the analysis could be more focused, and I have some concerns related to model performance at the two different resolutions, and how that translates into a potential recommendation for future research into health impacts. These aspects and others are described in detail below; addressing them will constitute minor revisions, after which this paper will be suitable for publication in ACP.

We thank the reviewers for their positive comments and encouragement.

**Major comments:**

**156: I understand the authors motivation here, to isolate the impact of model resolution from the impact of resolving differences in baseline mortalities. However, I disagree with their approach. But computing country-level AF and country-level baseline mortalities, the authors neglect any impact on mortality estimates that may come from sub-national variability in AF and baseline mortalities. It seems to me that a better (?) approach would be to map the O3 and PM2.5 concentrations from both the coarse and fine simulations to the same fine-scale resolution of the available population and baseline mortality rate information. This way they would have a consistent comparison that isolates the impact of the air quality model resolution, but their final estimates of mortality would be more accurate and more sensitive to differences in the air quality model resolutions. I'd suggest they at least consider this approach, which is just a postprocessing step and doesn't involve any more model simulations, to see if it makes a significant difference, or explain why it isn't the recommended approach.**

We thank the review for raising this point. Sub-national mortality rates that take into account variations in mortality within each country are not readily available across most European countries. However, we do account for sub-national variability in pollutant concentrations by applying population-weighting as described in Eq. (2) Section 2.3. If we were to calculate the differences in attributable fractions between the two resolutions at the model grid-level, their spatial distribution would be identical to that of the differences in warm season MDA8 $O_3$ and annual-mean $PM_{2.5}$ concentrations depicted in Fig. 5 as the AF is only dependent on the pollutant concentration and β (which is not available at the grid-cell nor country-level) following Eq. (1) Section 2.3. To illustrate this point we have calculated the difference in AF attributable to summer mean MDA8 $O_3$ at the grid-cell level (Fig. R4). Fig. R4 re-produces the spatial distributions of Fig. 5a in the manuscript with a scaling applied to the concentrations. For this reason we do not feel that calculating differences in AF at the grid-cell level between the two resolutions would add extra value to the manuscript. We have added the following text to section 4.3 to improve clarity and explain this point:

Page 14, line 442: *"If the AF was calculated for each model grid-cell rather than at the country level, the differences in AF for the two pollutants would have identical spatial distributions to the differences in warm season MDA8 $O_3$ and annual-mean $PM_{2.5}$ concentrations depicted in Fig. 5, as the AF is only dependent on the pollutant concentration and β (which is constant across all countries)."*

[Figure]

AFcoarse − AF finer (per grid cell) (%)

| | | | |
|---|---|---|---|
| 4.1 - 4.5 | 2.1 - 2.5 | 0.1 - 0.5 | -1.9 - -1.5 |
| 3.6 - 4.0 | 1.6 - 2.0 | -0.4 - 0.0 | -2.4 - -2.0 |
| 3.1 - 3.5 | 1.1 - 1.5 | -0.9 - -0.5 | -2.9 - -2.5 |
| 2.6 - 3.0 | 0.6 - 1.0 | -1.4 - -1.0 | -4.5 - -3.0 |

**Figure R4:** Differences in AF associated with long-term exposure to summer mean MDA8 $O_3$ between the two resolutions expressed as a percentage for each model grid-cell (AF$_{coarse}$ − AF$_{finer}$) using a threshold of 70 µg m$^{-3}$.

**When presenting the AF results, it would be interesting to know if the differences between the two scales of analysis are greater than the error bars in the AF estimates stemming from the uncertainty in the concentration response parameter (beta). In other words, when are the model-dependent differences significant, compared to the health-data uncertainties? See papers by Thompson et al. in this regard.**

We thank the review for pointing out this omission in our results. We have added the following text to the manuscript on this point in Sections 4.3 and 4.4.

Page 15 Line 458: "*When considering the uncertainty associated with the concentration-response coefficient used, the sign of the difference of AF between the two model resolutions is unaltered (Fig. 7b). Over the majority of the countries, the AF attributable to long-term exposure to MDA8 $O_3$ by the coarse resolution fall within the range of uncertainty as calculated by the finer resolution (Fig. 7a). However, over Finland and Ireland, the coarse mean estimates fall outside the uncertainty range estimates using the finer resolution (Fig. 7a).*"

Page 16, line 499, *"For a number of countries, the mean AF attributable to long-term exposure to PM$_{2.5}$ using the coarse resolution falls outside the uncertainty range of the finer estimates in particular over Iceland and Ireland (Fig. 8a)"*

Page 17, line 523, *"For differences in AF attributable to long-term exposure to summer mean MDA8 O$_3$ and annual mean PM$_{2.5}$ concentrations, the uncertainty associated with the concentration-response coefficient used does not alter the sign of the difference of AF between the two model resolutions (Fig. 7b and 8b). The uncertainty ranges for the PM$_{2.5}$ –related estimates show a greater variability between the two resolutions for more countries compared to MDA8 O$_3$-related AF estimates. Using the concentration-response coefficient in Jerrett et al. (2009), Thompson et al. (2014) find that the avoided mortalities due to difference in ozone concentrations between 2005 and 2014 at a 36 km model resolution are within the 95% uncertainty range associated with the concentration-response coefficient used compared to estimates at a resolution of 12 km and 4 km. These authors also find avoided mortalities associated with long-term effects of PM$_{2.5}$ exposure at 36 km to fall within estimates at the 12 km and 4 km resolution for three different concentration-response coefficients. Thus our results are in agreement for summer mean O$_3$ but less for annual mean PM$_{2.5}$"*

**It seems somewhat problematic, in terms of drawing conclusions, that the fine-scale simulated concentrations are, in many seasons, a poorer match to the observations than the coarse scale simulations, for both O3 and PM2.5. I strongly insist that the authors should present the statistical evaluation of biases in O3 during the warm sea-son and annual average PM2.5 in the main text, not the supplemental, as these are the scales most relevant to the focus of this work (health impacts). This is rather critical information that the reader shouldn't have to dig for.**

We thank the reviewer for this valid point and we agree that the statistical evaluation of biases in O$_3$ during the warm season and annual average PM$_{2.5}$ currently in Table S1 would fit better in the main text. Hence Table S1 has now been moved to Table 3.

**Overall, given these biases, would the authors recommend using the fine scale model over the coarse scale model for health impact analysis, especially for PM2.5 where the bias in the annual average concentration is higher at the finer scale? Or are there enough observations to say which is better at estimating exposure? This wasn't clear to me. I think this warrants some discussion, with conclusions in the abstract and conclusions.**

This is an interesting point. However, we cannot and do not wish to state if one model resolution is 'better' than the other in terms of health impact analysis, because this depends on many factors and the specific comparison. The main message we wish to convey is that the differences in pollutant concentrations between the two model resolutions, which in turn drive differences in AF, vary spatially and that we can quantify the ranges of these differences and explain why these occur. As already mentioned in the abstract and conclusion, for $PM_{2.5}$ concentrations the coarse resolution results in a lower bias in spring and autumn, while the finer resolution results in a lower bias in winter and summer. For annual $PM_{2.5}$ concentrations, the absolute difference in mean concentrations between the two resolutions is small (1.1 µg m$^{-3}$) hence, it is difficult to derive robust conclusions about which model resolution produces better results. This is not to say that the model performance in world regions where the ranges in $PM_{2.5}$ are greater might differ substantially between the two resolutions.

Moreover, as mentioned in the manuscript (Page 11 Line 335), we note that the available sites measuring $PM_{2.5}$ during our study period are not representative of the whole domain as measurements are lacking in the eastern part of Europe where we find higher annual mean $PM_{2.5}$ concentrations simulated at the coarse compared to finer configuration, particularly in summer and autumn (Fig 1 for site locations and Figs. 4 and 5 for seasonal/annual-mean $PM_{2.5}$ differences between the two resolution in the manuscript). In addition, whilst the bias in seasonal mean $O_3$ concentrations is higher for the finer resolution compared to the coarse resolution in most seasons, the concentrations at the finer resolution in some locations capture the high $NO_2$ and low $O_3$ concentrations associated with highly populated and thus polluted regions (Fig. 3). However, again we cannot definitively say which resolution more realistically estimates $O_3$ and $NO_2$ concentrations as available as site locations are not representative across the whole domain (Fig 1).

**Also, model bias relative to observations should be considered when discussing regional differences in modeled spatial resolution of population-weighted concentrations (section 4.2) – in other words, is one model resolution notably better in heavily populated areas? This is a critical question which I couldn't find a direct evaluation of, although all the pieces are available to make the comparison. The same comment applies to comparison of AF (section 4.3).**

We thank the reviewer for this comment and we agree this would be very insightful to add to the manuscript. However all the measurement locations in EMEP are urban background sites and not in densely populated areas, following the criteria for urban background site classification from Tørseth et al. (2012) and the EMEP manual (EMEP/CCC, 2001). For example, the minimum distance to emission and contamination sources from towns, power plants and major motorways is 50 km. Therefore these model to observation comparisons do not allow us to distinguish densely and non-densely populated locations. For clarification, we have added the following correction to the manuscript in Section 2.2 and Section 4.2.

Page 5 line 154: "*We note that all EMEP stations are classified based on a specific distance away from emission sources so as to be representative of larger areas. For example the minimum distance from large pollution sources such as towns and power plant is ~ 50 km (Torseth et al., 2012; EMEP/CCC, 2001).*"

Page 14 line 425: "*It would be insightful to examine these population-weighted results in relation to model-observation biases in densely populated areas. However, as outlined in Section 2.2, the available sites in the EMEP database are urban background stations which are required to be representative of a wide area and away from urban and industrial areas (EMEP/CCC,2001). Nonetheless we do note that in southern Europe, simulated summer mean MDA8 $O_3$ concentrations at the finer resolution are closer to observations than concentrations simulated at the coarse resolution. We find no consistent result for model biases in simulated annual mean $PM_{2.5}$ concentrations with respect to observations for the two model resolutions.*"

For AF estimates, comparison to observation is not possible as our estimates are calculated at the country level.

**426 - 437: Regarding comparison to studies in the US, I think an interesting conclusion is that the differences owing to model resolution is not something that is consistent in sign, spatially (or that could thus be easily corrected for without knowing the spatial dependence). This is self consistent with their own evaluation of the variability of the difference across regions within Europe. Still, one might hypothesize about additional factors that control these differences. Did the authors consider the speciation of the PM2.5**

**and how this might affect the differences between coarse and fine scale simulations? For example, both Punger and West (2013) and Li et al. (2015) note that the differences are more significant for primary anthropogenic PM (e.g., BC) than secondary anthropogenic PM or primary natural PM. I contrast, Thompson 2014 noted the biggest impact of resolution going from 36 km to 4 km was for secondary PM. I didn't see PM2.5 speciation discussed anywhere in the present work.**

We thank the reviewer for their insights. We had analysed differences in $PM_{2.5}$ components between the two resolutions. However we found no substantial differences. We have added the following text in Section 4.4 to highlight previous findings and our results.

Page 17 Line 512: "*In contrast, Thompson et al. (2014) find that health benefits associated with changes in PM$_{2.5}$ concentrations between 2005 and 2014 in the U.S., were not sensitive to resolution. Both Punger and West (2013) and Li et al. (2015) find that differences in PM$_{2.5}$ are mainly attributable to primary anthropogenic PM, while Thompson et al. (2014) attribute the greatest differences (between 36 km and 4 km resolutions) to secondary PM. However, in our study no substantial differences in PM$_{2.5}$ components between the two resolutions were found.*"

**Minor comments:**
**1.13: Given that it is a regional modeling study, would make more sense to phrase as "at resolutions typical of global (_140 km) and regional (_50km) models." Throughout, it would make more sense to me if the results were referred to as "coarse" vs "fine" rather than "global" vs "regional", all results are regional and this could be misleading to someone just glancing at the figures. Further, most regional models these days are more like 12 km scale or finer.**
Yes this is a good point and we agree. Therefore throughout the manuscript, the words "*global*" and "*regional*" have been changed to "*coarse*" and "*finer*". For further clarity as noted to our responses to Reviewer 1, we have added the words 'for Europe' to the title of the manuscript.

**1.15: The differences seem a bit more modest, all less than 30% and most less than 10%. Not sure if "strong" is the right word.**
We have now removed the word "*strong*" from lines 16, 293 and 539 to address this point.

**28-33: Readers not familiar with AF may think these numbers are very small – the authors might wish instead (or additionally) to present the amount by which AF is changing owing to model resolution (i.e. a factor of two to three). So consider not the changes in total mortality (which is small, 5%) but instead the % changes in pollution attributed mortality. I think the authors should also state the differences in the total over the entire domain, rather than just the range across regions, even if the total benefits from some fortuitous cancelation of under and over estimates.**

We agree that some readers may think these numbers are small. Therefore when stating the ranges of differences in AF between the two resolutions, we have added the following text to sections 4.3 and 4.4 to highlight these factors when discussing individual countries with the largest percentage changes. We left the numbers in the abstract as is as the percentage values for differences in AF are of the same order of magnitude as the differences in concentrations between the two resolutions given above.

Page 15 Line 455: *"In Poland and Portugal the estimated AF at the finer resolution is 1.4 times and 0.7 times respectively that estimated at the coarse resolution."*

Page 16 Line 495: *"For Cyprus and Croatia, using the finer resolution results in an estimated AF that is 1.5 and 0.7 times that estimated using the coarse resolution."*

Differences in the AF total over the entire domain are outweighed by cancelation of under and over estimates leading to a very small average difference and since we wish to highlight regional differences we feel that this addition would not add value to the manuscript.

**52-62: I'm not sure how "strong" the effects are that are being discussed in this paragraph – can the authors be more quantitative when describing previous works? The following paragraph on PM2.5 is much better in this regards.**
We thank the reviewer for this suggestion. We have now updated these lines in Section 1 to include numerical values from the Stock et al. (2014) paper as indicated in the text below. However Valari and Menut (2008) state "the model is more sensitive to changes in the resolution of emissions than in meteorological input" with no quoted numerical values and only refer to Fig. 7. Hence we did not feel we could provide a further quantitative analysis since the authors did not explicitly state this.

Page 2 lines 60-65: "*Furthermore, Stock et al. (2014) found the impact of spatial resolution (150km vs. 40km) on simulated $O_3$ concentrations to vary with season across Europe. In winter, higher $NO_x$ concentrations produced more pronounced titration effects on $O_3$ at 40 km resolution with a mean bias error (MBE) of 3.2%, leading to lower $O_3$ concentrations than at 150 km resolution (MBE = 14.4%). In summer, although similar results were found for $O_3$ concentrations simulated at the coarse (MBE = 29.7%) and fine resolution (MBE = 32.8%) simulated boundary layer height was suggested to be largely responsible for the spatial differences in $O_3$ concentrations at the two resolutions.*"

**68: The reference is Li et al. (2015). not 2016. The authors refer to the paper both ways in the text but only include an incorrect reference to 2016 in the bibliography.**

We thank the review for pointing out this mistake. We have now corrected this reference to "*Li et al. (2015)*" both in the text and in the reference list.

**78: The authors should also consider the results of Thomson and Selin (2012), which found there were some differences between O3-related premature deaths at the 36 km scale and finer (24, 12, and 4 km) scales, although these tended to lie within the range of uncertainty of the health impact estimates. Further, they should discuss the O3 health impact results from Thompson 2014, which were found to be more sensitive to resolution than PM2.5 health impacts.**

We thank the reviewer for this suggestion. We have now added the following text to the revised manuscript in Section 1 to address both points.

Page 3 line 79-84: "*Thompson et al. (2014) also found that especially in urban areas, the human health impacts associated with differences in $O_3$ between 2005 and 2014 calculated using a coarse resolution model (36 km) were on average two times greater than those estimated using finer scale resolutions (12 km and 4 km). In addition,* Thompson and Selin (2012) *found that the estimated avoided $O_3$-related mortalities between a 2006 base case and a 2018 control policy scenario at a 36 km resolution were higher compared to estimates at finer resolutions (12 km , 4 km and 2 km). However, their health estimates at the 36 km resolution fall within the range of values obtained using concentrations simulated at the finer resolutions used .*"

**Table 1: The placement of the "difference" row is confusing. It is the difference in the model simulated mean, and should be labeled as such and more clearly located directly below the row reporting the mean, not the rows reporting NMB and SD. The choice of significant figures for the difference also seems odd. For example, why the DJF mean difference is reported as 10 rather than 9.6 isn't clear, when other numbers are resolved to the tenth of a ug/m3. Same comments apply to Table 2. Annual average PM2.5 and warm-season (April - Sept) O3 should be added to these tables, not put in the SI.**

We agree it is more intuitive to have the difference row below the mean concentration. Tables 1 and 2 have been updated accordingly and percentages have been rounded to 1 d.p. to be consistent with the rest of the values quoted. We have double checked the figures and found that the difference in magnitude of the mean spring $PM_{2.5}$ concentrations between the two resolutions was incorrect (the sign remains unchanged). This has been updated from -27% to -5.5% in Table 2, Abstract, main text and Conclusions.